# Partial observation can induce mechanistic mismatches in data-constrained models of neural dynamics

**William Qian**[1,2]**, Jacob A. Zavatone-Veth**[3,4,5]**, Benjamin S. Ruben**[1]**,  Cengiz Pehlevan**[2,3,4]

[1]Biophysics Graduate Program,
[2]Kempner Institute for the Study of Natural and Artificial Intelligence,
[3]John A. Paulson School of Engineering and Applied Sciences,
[4]Center for Brain Science, [5]Department of Physics,
Harvard University
Cambridge, MA 02138
`jzavatoneveth@fas.harvard.edu, cpehlevan@seas.harvard.edu`

## Abstract

One of the central goals of neuroscience is to gain a mechanistic understanding of how the dynamics of neural circuits give rise to their observed function. A popular approach towards this end is to train recurrent neural networks (RNNs) to reproduce experimental recordings of neural activity. These trained RNNs are then treated as surrogate models of biological neural circuits, whose properties can be dissected via dynamical systems analysis. How reliable are the mechanistic insights derived from this procedure? While recent advances in population-level recording technologies have allowed simultaneous recording of up to tens of thousands of neurons, this represents only a tiny fraction of most cortical circuits. Here we show that observing only a subset of neurons in a circuit can create mechanistic mismatches between a simulated teacher network and a data-constrained student, even when the two networks have matching single-unit dynamics. In particular, partial observation of models of low-dimensional cortical dynamics based on functionally feedforward or low-rank connectivity can lead to surrogate models with spurious attractor structure. Our results illustrate the challenges inherent in accurately uncovering neural mechanisms from single-trial data, and suggest the need for new methods of validating data-constrained models for neural dynamics.

## 1   Introduction

In recent years, advances in recording techniques have brought forth a deluge of neural data. Simultaneous measurements of the activity of hundreds to thousands of neurons can now be obtained at high spatiotemporal resolution [1–3]. These methods are increasingly deployed to perform longitudinal recordings in animals executing quasi-naturalistic behaviors or complex tasks [2–7], meaning that one may not have recourse to repeatable trial structure when analyzing these data [8]. A critical question for contemporary systems neuroscience then arises: How can mechanistic insights about the neural dynamics underlying animal behavior be extracted from large-scale recordings [3, 5, 7, 9, 10]?

Given access to only a single, non-repeatable measurement of neural activity, a natural question is whether one could construct a reliable *in silico* surrogate model for the dynamics of the measured neural circuit. As this surrogate model would not be subject to measurement limitations, it could be used to generate hypotheses about the corresponding biological neural populations, and to simulate how such populations might behave under various external inputs or perturbations. A natural approach to constructing a surrogate model is to optimize a recurrent neural network (RNN) to mimic the

38th Conference on Neural Information Processing Systems (NeurIPS 2024).

recorded neural activity. In recent years, this approach has gained broad popularity, and has been applied to data from many species and recording modalities [9, 11–20].

Data-driven models of neural dynamics are constructed under a number of less-than-ideal conditions. First, in single-trial settings, one has access to measurements of dynamics only in a restricted condition or set of conditions, so it may be impossible to observe how changes in input or in internal state affect the circuit of interest [8]. Second, the system of interest is only partially observed: one can only usually record from a subset of the neurons in a given circuit, and certainly cannot simultaneously record all of their inputs [7]. Third, measurements of neural activity are noisy, and may be biased to capture only certain components of neural activity. For example, intrinsic indicator dynamics mean that calcium imaging may effectively low-pass filter neural activity, even if one is only interested in firing rates and not precise spike times [21, 22]. Fourth, it can be challenging to account for the presence and structure of intrinsic neuronal noise. Finally, there will always be a significant mismatch between the single-unit dynamics of a model and biology, as models abstract away or ignore biophysical details to enable efficient optimization and simulation [17, 18].

Even in the unrealistic scenario where the activity of every relevant neuron is recorded, exactly inferring synaptic weights from dynamical measurements alone is extremely challenging [23]. A more modest hope is that data-constrained models should be able to capture the mechanistic dynamical properties of ground-truth circuits at a qualitative level—that is, to recapitulate slow time scales, unstable directions, oscillatory dynamics, and attractors [9, 12–14, 17, 19, 24, 25]. Given a data-constrained model, one can identify attractor properties using dynamical systems analysis [13, 26, 27]. These macroscopic dynamical properties are of substantial neuroscientific interest, as low-dimensional attractors are believed to underlie observed neural activity across a variety of neural circuits and tasks [24]. In particular, line attractors—sets of stable fixed points organized along lines in neural activity space—have been proposed to underlie cognitive functions requiring short-term or working memory, including sensory integration, decision making, and even aggressive behavior [12, 13, 24, 26, 28, 29]. Importantly, low-dimensional structures in data should be relatively robustly detectable even under partial observation [7], so there is reason to be optimistic that data-constrained models could correctly recover line attractor dynamics. Indeed, several recent papers have used data-constrained models with low-dimensional latent RNN dynamics to propose that line attractors underlie the accumulation of internal drives and of external reward [12, 13, 30, 31].

However, despite some positive examples [14, 19], previous works have not mapped out how partial observation affects whether data-driven modeling can accurately recover low-dimensional attractor structure. To address this question, in this paper, we consider a teacher-student setup in which activity from one RNN is imitated by another, and show that partial observation can induce mechanistic mismatches even under relatively ideal conditions where the input to a circuit is either perfectly known or white noise, and where the single-unit dynamics of the student match the teacher. Our primary contributions are as follows:

- In §2, we begin with a motivating example: we show that data-constrained modeling fails to distinguish between two mechanistically-distinct models in a stimulus-integration task. Both a line attractor network [28] and a functionally feedforward chain [32] are identified as line attractors.
- We then turn to the analytically-tractable setting of noise-driven linear RNNs (§3). In §3.1, we show that when the teacher is an approximate line attractor, the student will recover this structure. In contrast, when the teacher connectivity is non-normal, the student may learn spurious approximate attractor structure. We illustrate this with two biologically-motivated examples: functionally-feedforward integrators (§3.2), and networks with low-rank connectivity (§3.3).
- Then, in §4, we explore how these insights generalize beyond the linear setting. Focusing on the example of nonlinear low-rank networks, we show that partial observation once again can induce overestimation of eigenvalue magnitude. Here, though, this can result in spurious attractor structure including additional stable fixed points and limit cycles.

Our results begin to illuminate the inductive biases of data-constrained RNNs trained under partial observation towards particular mechanisms of generating long timescales. They suggest that caution is warranted in inferring mechanism from data-constrained models, and underscore the primacy of direct activity perturbations for validating putative attractor dynamics [25].

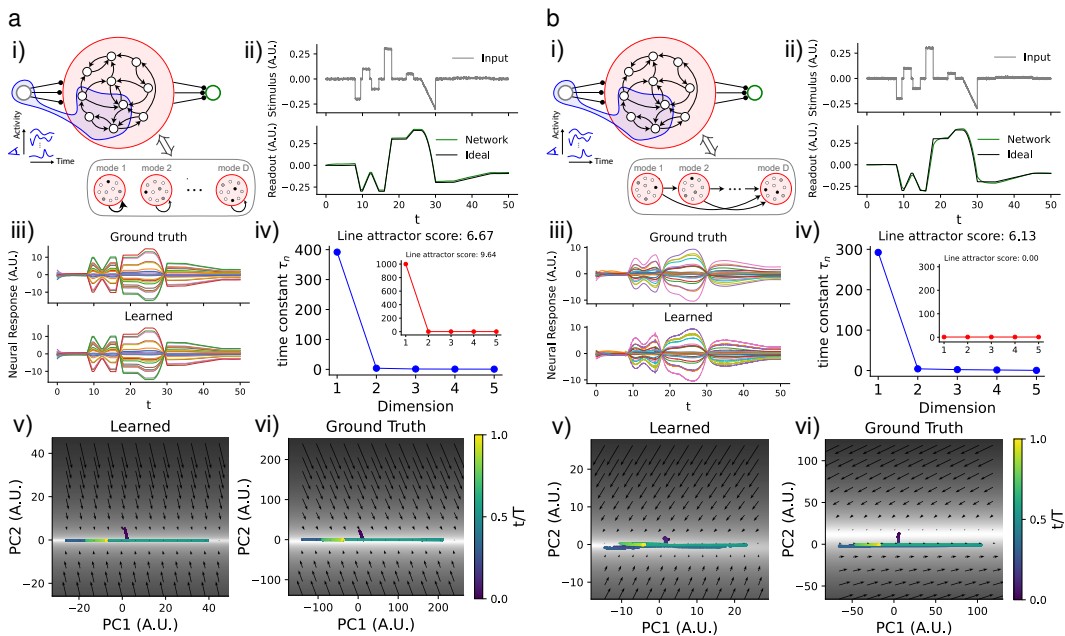

Figure 1: Data-constrained models fail to distinguish between mechanistically different sensory integration circuits. **a.** Recovery of a line attractor through data-constrained modeling. i). Schematic of integrator network, showing the subsampled neurons (blue), and its interpretation as a set of independent self-excitatory modes. ii). Input signal (*top*) and its integral (*bottom*) as estimated by the network (*green*) and computed exactly (*black*). iii). Example activity traces from the true network (*top*) and an LDS fit to observations of 5% of its neurons (*bottom*). The agreement is excellent. iv). Spectrum of time constants for the data-constrained LDS model (*main figure*) and for the top five time constants of the true circuit (*inset*). Both show a single large time constant, indicating approximate line attractor dynamics, though the data-constrained model underestimates that of the true network. v-vi). Flow field in the space of the top two principal components of activity for the LDS model (v) and line attractor network (vi). Shading indicates the magnitude of the flow, while arrows indicate its direction. Observed activity is shown by dots colored by their time. The learned flow field shows good qualitative agreement with the ground truth; both networks have a slow line along which the observed activity is driven. **b.** As in **a**, but for a functionally-feedforward integrator circuit. As diagrammed in (i), this network can be thought of as a set of non-self-exciting modes which are connected in a feedforward chain. Though this network solves the integration task (ii) and the LDS fit is good (iii), the LDS identifies a single long time constant that is not present in the true dynamics (iv). The learned (v) and ground-truth (vi) flow fields correspondingly do not match, with the activity lying off the slow line of the true dynamics. See Appendix G for detailed experimental methods.

## 2 A motivating example: data-constrained modeling of integrator circuits

The circuit basis for temporal integration of scalar sensory inputs is a longstanding question in systems neuroscience [12, 24, 28, 29, 32–41]. Though many models for integrator circuits have been proposed [28, 32, 37, 40, 42], two linear RNN models are perhaps the most prominent: the line attractor [28, 40], and the feedforward chain [32, 37]. Both of these models have extremely simple dynamics

$$\tau \dot{\mathbf{z}} = -\mathbf{z} + J\mathbf{z} + \mathbf{b}u$$

for state $\mathbf{z} \in \mathbb{R}^D$, recurrent weights $J \in \mathbb{R}^{D \times D}$, and input $u(t) \in \mathbb{R}$ encoded through $\mathbf{b} \in \mathbb{R}^D$. However, they posit structurally distinct mechanisms for how memories can be maintained beyond the single-unit time constant $\tau$. In classic line attractor networks, the recurrent weights are chosen to be symmetric, and one eigenvalue of $J$ is tuned to be precisely equal to one, with the rest being less than one. Then, by choosing the input weights $\mathbf{b}$ to align with the corresponding eigenvector, one obtains a perfect integrator of the signal $u(t)$ [28] (App. A). However, this model suffers from a substantial fine-tuning problem: slight mis-tuning of the weights causes exponentially large deviations from perfect integration [28, 32, 37]. In contrast, a functionally feedforward chain maintains a memory by iteratively passing signals from one mode of activity to the next (Fig. 1; App. A) [32, 37]. Most

simply, a literal feedforward chain has connectivity $J_{ij} = \delta_{i+1,j}$. However, one can encode modes in distributed patterns of neural activity rather than single neurons, so that this structure is not obviously apparent in recordings (see App. A for details). Such networks are more robust to mistuning of synaptic strengths than line attractor networks, but they can only sustain a memory over $\mathcal{O}(\tau D)$ time. Importantly, the dynamics of such a network are highly non-normal; the recurrent connectivity matrix $J$ has all eigenvalues equal to zero. Here, inspired by [36], we add skip connections from each mode to the last mode in the chain (Fig. 1; App. A). This guarantees that, like the line attractor network, the activity produced by the functionally feedforward network is approximately low-dimensional.

Given the simplicity and ubiquity of these models, we first asked whether data-constrained modeling could robustly distinguish between them. We constructed a model sensory integration task, which networks of both architectures could effectively solve (Fig. 1). Using standard variational inference methods [13, 18], we fit recordings of 5% of the neurons from each network with a latent linear dynamical system (LDS), which models the neural activity as a linear projection of a low-dimensional latent RNN [18] (App. G).[1] These models explicitly encode the prior belief that population activity is low-dimensional. In this case, we used 5 latent dimensions.

Though the data-constrained models do an excellent job of capturing the activity recorded from both the line attractor and the feedforward chain, analyzing the latent dynamics matrices reveals that both networks are interpreted as approximate line attractors (Fig. 1). In particular, the spectrum of eigenvalues $\hat{\lambda}_i$ of each LDS dynamics matrix induces a spectrum of decay time constants $\hat{\tau}_i = \tau/|1 - \Re\hat{\lambda}_i|$ (in continuous time; see App. A and B.3) [13, 27]. Previous works have identified networks with large gaps between the top two timescales as approximate line attractors [13, 27]. As a simple metric, Nair et al. [13] defined a "line attractor score" $\log_2(\hat{\tau}_1/\hat{\tau}_2)$, and interpreted scores greater than 1 as indicative of approximate line attractors. The LDS models fitted to these mechanistically different integrator circuits each have a single slow direction, with a line attractor score in excess of 6 (Fig. 1). However, visualising the flow fields of the ground truth and data-constrained models shows that the dynamics of the line attractor are qualitatively recovered well, while the model fit to recordings of the feedforward chain shows a strong mismatch as it discovers a spurious line attractor (Fig. 1). Therefore, data-driven modeling fails to distinguish circuit hypotheses for this simple task.

An intuitive explanation for why the data-driven model fails to mechanistically reproduce the functionally-feedforward chain is immediate: if the single-unit time constants are fixed and the data-constrained model has fewer neurons, it cannot realize a feedforward chain with sufficiently long memory [32, 37]. The only way to manufacture long memory timescales with a small number of latent neurons is through large eigenvalues. This is potentially a fundamental obstacle to the ability of latent space models to recover neural mechanisms; we will return to this point in the Discussion.

## 3 A tractable model setting: noise-driven linear networks

Motivated by the observations of the previous section, we now seek a setting in which we can analytically study the structure of the student RNN's weight matrix. Whereas in §2 we assumed the teacher networks were driven by a known low-dimensional signal, here we consider the case in which the teacher and student are driven by isotropic Gaussian noise. This is an optimistic assumption, as it means that the teacher network will explore all directions in its phase space evenly over the course of a single long trial [37].

Concretely, we consider a teacher-student setup in which both networks are rate-based linear RNNs driven by isotropic Gaussian noise, or, equivalently, their activity evolves according to multivariate Ornstein–Uhlenbeck (OU) processes [43]. The teacher has $D$ neurons and a recurrent weight matrix $B$, such that the dynamics of its firing rate vector $\mathbf{z}(t) \in \mathbb{R}^D$ is

$$\tau\dot{\mathbf{z}} = -\mathbf{z} + B\mathbf{z} + \boldsymbol{\xi}(t)$$

where $\boldsymbol{\xi}(t)$ is white Gaussian noise. The student's dynamics are identical, except that it has $d$ neurons, recurrent weights $A$, and driving noise $\boldsymbol{\eta}(t)$, such that its rate $\mathbf{x}(t) \in \mathbb{R}^d$ evolves as

$$\tau\dot{\mathbf{x}} = -\mathbf{x} + A\mathbf{x} + \boldsymbol{\eta}(t).$$

Then, the task is to estimate the student's dynamics matrix $A$ given access only to partial observations of the teacher's activity. For simplicity, we assume that we observe the first $d$ neurons of the teacher

---

[1]All code is available at https://github.com/wqian0/DataConstrainedRNNs/.

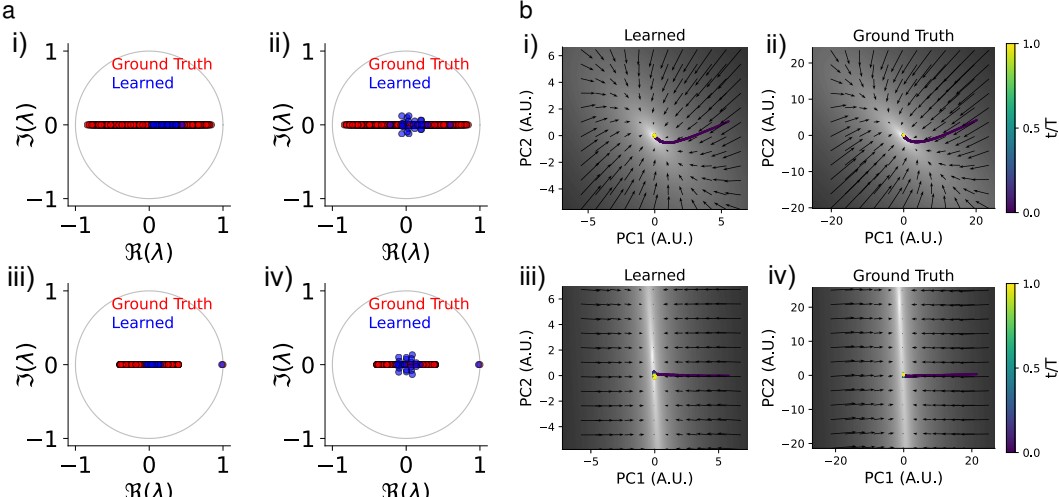

Figure 2: Partial observation of symmetric teacher networks does not lead to spurious attractor dynamics in a data-constrained student network. **a.** Ground truth teacher (red) and learned student (blue) dynamics matrix eigenvalues. (i),(ii): symmetric teacher without attractor structure. (iii),(iv): symmetric teacher that is an approximate line attractor. (i),(iii): for infinite observation time. (ii),(iv): for a finite observation time window. **b.** Flow fields of learned (student) and ground truth (teacher) networks for a finite observation window. (i),(ii): symmetric teacher without attractor structure. (iii),(iv): symmetric teacher that is an approximate line attractor. All plots correspond to 5% partial observation. See Appendix G for detailed experimental methods.

network for time $T$, i.e., we observe

$$\mathbf{x}^{\text{obs}}(t) = P\mathbf{z}(t) \quad \text{for} \quad t \in [0, T] \quad \text{and} \quad P = (I_d, \quad 0_{d \times (D-d)}).$$

Assuming an isotropic Gaussian prior $A_{ij} \sim_{\text{i.i.d.}} \mathcal{N}(0, 1/(\rho T))$ scaled such that the long-time limit is well-defined, we show in Appendix B that the maximum *a posteriori* (MAP) estimate of $A$ can be computed explicitly in terms of empirical covariances of $\mathbf{x}^{\text{obs}}(t)$ [44–47]. To make the problem analytically tractable, we focus on the limit $T \to \infty$, where these covariances can be computed using classical results on stationary states of OU processes [43, 48]. We assume that the eigenvalues of the teacher's weight matrix $B$ have real part strictly less than one, such that it admits a stable stationary state with covariance $S = \lim_{t \to \infty} \mathbb{E}[\mathbf{z}(t)\mathbf{z}(t)^\top]$. Then, in the limit of a long observation window $T \to \infty$, the MAP estimate of the student's dynamics matrix can be written in terms of the stationary covariance $S$ as (App. B)

$$\hat{A}_\infty = PBSP^\top (PSP^\top + \rho I_d)^{-1}.$$

This result is stated in continuous time; we also give the corresponding result for discretized dynamics in Appendix B. In the fully-observed case, the zero-ridge limit of the MAP recovers the teacher dynamics matrix, i.e., $\lim_{\rho \downarrow 0} \hat{A}_\infty|_{d=D} = B$. The stability condition means that we can consider at best approximate line attractors with arbitrarily large but not infinite time constants, but this is not a substantial limitation [28, 32, 37].

To determine when this data-driven modeling approach recovers the mechanistic structure of the teacher, our task is then to analyze the spectrum of $\hat{A}_\infty$ for various choices of $B$, as for linear networks the eigenspectrum fully determines the (approximate) attractor structure [28]. Concretely, letting $\lambda_i$ and $\hat{\lambda}_i$ be the eigenvalues of the teacher and student dynamics matrices, respectively, we want to compare the resulting spectra of timescales $\tau_i = \tau/|1 - \Re\lambda_i|$ and $\hat{\tau}_i = \tau/|1 - \Re\hat{\lambda}_i|$. We are primarily interested in whether the existence or non-existence of slow timescales can be accurately recovered.

## 3.1 Normal dynamics

We begin by considering teacher networks with normal connectivity matrices ($BB^\top = B^\top B$). This includes attractor networks like the idealized line attractor, which have symmetric connectivity ($B = B^\top$), and when driven by noise have an equilibrium stationary state [43, 48]. As such matrices

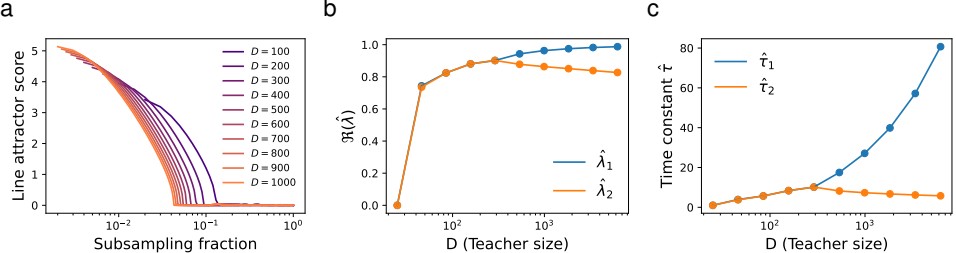

Figure 3: Heavily subsampling a feedforward chain leads to line-attractor-like student dynamics. **a.** Line attractor score as a function of subsampling fraction $d/D$ for teacher networks of varying sizes $D$. **b.** Real parts of the top two eigenvalues of a $d = 25$ student's dynamics matrix for varying teacher network size $D$. **c.** As in **b.**, but showing the time constants corresponding to the top two eigenvalues. Beyond a threshold value of $D$, the separation increases rapidly. Thus, the student shows two mechanistic mismatches: First, it learns a dynamics matrix with non-vanishing eigenvalues. Second, at sufficiently low subsampling fraction the top two eigenvalues are separated by a substantial gap, yielding line-attractor-like dynamics. See Appendix G for detailed experimental methods.

have orthogonal eigenspaces, the dynamics of a normal teacher network can be viewed as a set of non-interacting modes with decay timescales determined by the real parts of the eigenvalues (App. A).

For such teachers, we show in Appendix C that partial observation does not lead to overestimation of timescales under MAP inference. Ordering the eigenvalues of $B$ in descending order of their real parts as $1 > \Re(\lambda_1) \geq \Re(\lambda_2) \geq \cdots \geq \Re(\lambda_D)$, the eigenvalues $\hat{\lambda}_i$ of the student's dynamics matrix $\hat{A}_\infty$ satisfy $\Re(\lambda_1) \geq \Re(\hat{\lambda}_i) \geq \Re(\lambda_D)$ for all $1 \leq i \leq d$. However, this positive recovery result does not exclude the possibility that the spectrum of the student's dynamics matrix will have qualitatively distinct gap structure, which would lead to incorrect inference of approximate attractor mechanisms.

In the special case of an ideal line attractor, this does not happen: if the teacher is a symmetric approximate line attractor, then the student will be as well. Concretely, suppose that $B$ is symmetric, with eigenvalues satisfying $\lambda_1 = 1 - \varepsilon$, $\varepsilon \ll 1$, and $\lambda_i \ll 1$ for $i \geq 2$, and that the eigenvector $\mathbf{u}_1$ corresponding to the leading eigenvalue (the direction of the approximate line attractor) is randomly oriented or delocalized. Then, the eigenvalues of the student dynamics matrix satisfy $\hat{\lambda}_1 \geq \lambda_1 - \mathcal{O}(\varepsilon D/d)$ and $\hat{\lambda}_2 \leq \lambda_2$ (App. C.3). This implies that approximate line attractors can be recovered even under heavy partial observation so long as the deviation $\varepsilon$ of the teacher dynamics from a perfect line attractor is small. In Figure 2, we illustrate this successful recovery, and show that it is not qualitatively affected even if the observation time is finite. This successful recovery is consistent with what we found in the driven setting in Figure 1.

### 3.2 Non-normal dynamics: Feedforward amplification

Our results for normal teacher dynamics in §3.1 show that the student can correctly recover line attractor dynamics, matching our motivating observation in Figure 1. However, we recall that we found that a non-normal network performing integration through feedforward amplification was incorrectly recognized as also being a line attractor. While it is challenging to analyze general non-normal teacher matrices in the noise-driven setting [43, 48], we can show that this mismatch again emerges for feedforward chains. In particular, we show in Appendix D that the dynamics of a student of fixed size approach that of a line attractor as teacher size increases. Assume that the teacher is a perfect feedforward chain with connectivity $B_{ij} = \delta_{i+1,j}$. Then, as $D \to \infty$ for fixed $d$, the student dynamics matrix $\hat{A}_\infty$ in the limit of long observation time and vanishing regularization approaches $\delta_{i+1,j} + \delta_{id}\delta_{ij}$, hence its leading eigenvalue approaches 1, while the others tend to zero (App. D).

As illustrated in Figure 2, the qualitative conclusion that partial observation leads to timescale overestimation does not change even when the observation time is finite. We remark that the fact that the student becomes closer and closer to a line attractor as $D$ increases is consistent with the intuitive argument given at the end of Section 2: if the number of observed neurons is fixed and small, the only way for the student network to capture the long integration window of the feedforward chain is through tuning its eigenvalues to create long timescales. In Figures 3 and G.3, we substantiate this intuition by showing how the estimated timescales depend on the size of the teacher network relative to the student.

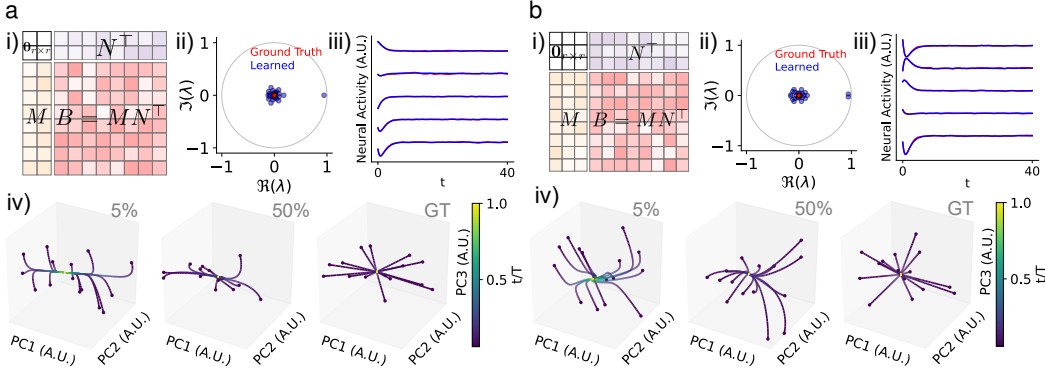

Figure 4: Spurious slow directions in data-constrained student models for low-rank teacher dynamics. **a.** Learning from a rank-2 teacher. i). Schematic of teacher weights. ii). Ground truth teacher (red) and learned student (blue) dynamics matrix eigenvalues at 5% subsampling. Note the presence of a single learned outlier eigenvalue with real part near 1. iii). Activity traces for the teacher (red) and student (blue) networks. iv). Example student network dynamics for 5% and 50% subsampling compared to the ground truth (GT). Here, points along the trajectory are colored by their time. The student dynamics rapidly converge to a line and then decay slowly towards the origin, consistent with the outlier eigenvalue observed in (ii). **b.** As in **a**, but for a rank-3 teacher network. Correspondingly, the student learns two outlier eigenvalues, and two slow directions. See Appendix G for detailed experimental methods.

### 3.3 Low-rank non-normal dynamics

As a second neuroscience-inspired example of non-normal teacher dynamics, we consider low-rank connectivity. In recent years, low-rank RNNs have emerged as popular models for cortical dynamics [15, 16, 25, 49, 50]. Importantly for our purposes, they yield low-dimensional population activity, and hence are again a relatively ideal scenario for data-constrained modeling under partial observation [7, 50]. However, we find that connectivity that is both non-normal and low-rank can also give rise to severe timescale overestimation in the student network.

As a particularly simple example of low-rank teacher dynamics, we consider the case in which $B = MN^\top$ is rank $r \ll D$, with $M, N \in \mathbb{R}^{D \times r}$ having null overlap $M^\top N = \mathbf{0}_{r \times r}$ and orthogonal columns $M^\top M = N^\top N = \gamma^2 I_r$. Then, $B$ is a non-normal matrix with all-zero eigenvalues. When $\gamma \gg 1$, the stationary covariance $S$ of the teacher network's activity will have precisely $r$ large eigenvalues of order $\gamma^4$, separated from a bulk of eigenvalues that are of $\mathcal{O}(1)$ with respect to $\gamma$ (App. E). In this large-$\gamma$ regime where the teacher's activity is approximately low-dimensional, the student's learned dynamics matrix has $r$ eigenvalues approaching 1, with the rest approaching zero (App. E). Therefore, the student learns an $r$-dimensional hyperplane attractor. Importantly, this can occur when $\gamma^2$ is chosen such that $B$ has order-1 elements. We show how this effect depends on subsampling fraction in G.4.

In simulations, we observe a finite observation time effect whereby only $r - 1$ of the learned eigenvalues are near 1 when process noise is small (Fig. G.1). Consequently, fitting a student network to a non-normal teacher with null overlap connectivity of rank $r$ as described above can result in the spurious discovery of approximate $(r - 1)$-dimensional hyperplane attractors. We illustrate this explicitly for the cases $r = 2$ and $r = 3$, where observing only 5% of the neurons in the teacher network leads to the spurious discovery of approximate line attractor and plane attractor dynamics, respectively, despite nearly perfectly recapitulating the observed activity (Fig. 4). Consistent with these observations, we show that latent LDS models fit to the same teacher activity via more sophisticated variational inference methods also learn a few vastly enlarged timescales (Fig. G.5).

## 4 Mismatched attractor structure in data-constrained nonlinear networks

Though the linear networks studied in §3 are analytically tractable, they are of course inherently limited in the types of attractor dynamics they can display. We therefore asked which qualitative insights from the linear setting carry over to nonlinear networks where one allows for a nonlinear

firing rate transfer function $\phi$. That is, we again consider a student-teacher setup, but now the teacher and student dynamics are $\tau\dot{\mathbf{z}} = -\mathbf{z} + B\phi(\mathbf{z}) + \boldsymbol{\xi}(t)$ and $\tau\dot{\mathbf{x}} = -\mathbf{x} + A\phi(\mathbf{x}) + \boldsymbol{\eta}(t)$, respectively (App. B). Our focus is again on low-rank networks, both for their usage as models for cortical processing and for the fact that their approximately low-dimensional activity makes them a natural candidate for data-driven modeling under partial observation [15, 16, 49].

While it is less straightforward to relate attractor dynamics to the eigenspectrum of $B$ in the nonlinear setting, one can still use spectral information to gain insight into the dynamics near the trivial fixed point at the origin. Specializing to $\phi(z) = \tanh(z)$, the Jacobian of the teacher dynamics at the origin is simply $-I_D + B$, and that of the student is analogously $-I_d + A$. Therefore, connectivity matrix eigenvalues of real part greater than $1$ would imply that the fixed point at the origin is unstable, and thus the network must support other dynamical behavior (e.g, other stable fixed points, limit cycles, and/or chaos). In the linear case, eigenvalues of real part greater than $1$ were never spuriously discovered, as that would yield exponentially divergent activity. However, with a saturating nonlinearity, this extreme eigenvalue overestimation is no longer pathological. Indeed, when we infer the weights of a student network using MAP estimation, we find that at small subsampling fractions the student can learn eigenvalues of real part greater than one from a teacher with no such eigenvalues if the teacher connectivity is non-normal. Strikingly, this can lead to the discovery of spurious limit cycles (Fig. 5a) and fixed points (Fig. G.6).

At this point, one might ask whether the eigenvalue overestimation phenomenon we have observed is an artefact of the estimation methods (thus far, MAP and LDS variational inference) on which we have focused. The conceptual argument given at the end of §2 suggests that this should hold more broadly for low-dimensional student networks learning from partial observations of high-dimensional teachers, but this is a heuristic argument, not a rigorous test. We therefore applied several other commonly-used inference methods [14, 51, 52] to fit student dynamics in the low-rank nonlinear teacher setting. All of the methods produced mismatched attractor structure, with many showing a propensity to overestimate eigenvalues, yielding spurious limit cycles (Fig. 5). We show that these effects persist for more general teacher weight matrices and for student networks with additional hidden units to account for unobserved neurons in Fig. G.2.

## 5   Discussion

In this paper, we have shown partial observation can lead data-constrained models to incorrectly identify the mechanistic basis for slow recorded neural dynamics. We found that, while attractor-like networks can be faithfully recovered even when only a small fraction of neurons are recorded, data-constrained models can learn spurious attractor structure from non-normal transient dynamics.

As noted in §2, an intuitive explanation of our results for linear networks is that low-dimensional dynamical systems are limited in the longest timescales they could generate through functionally feedforward integration, and thus are inherently biased towards line-attractor-like mechanisms when fit to observations of slow dynamics. Though our focus has been on partial observation as a driver for this dimensional restriction, most approaches to data-constrained modeling with latent dynamics explicitly bias model selection towards smaller latent spaces. In particular, it is standard to select the smallest latent space dimension that captures more than a certain threshold fraction of the variance in the data [13, 18]. This will necessarily favor approximate-attractor-like solutions. Indeed, if one applied such a model selection procedure to the integrator models studied in Figure 1, one would select at most a two-dimensional latent space (see Figure G.8), and thus fall victim to the failure mode noted there. This bias in model selection procedures illustrates a wider issue: benchmarking and model selection based on explained variance for a restricted set of measured dynamics alone are not necessarily sufficient to diagnose mechanistic mismatches [23, 53]. It highlights a tension between the desire to recapitulate mechanism and our intuitive conception of low dimensionality as a signature of model parsimony.

Here, we have focused on the setting in which one measures dynamics over a single trial for inputs that are either fully known or white noise. Previous works have shown how failure to account for unobserved inputs can lead to incorrect eigenvalue estimation [19]. Moreover, most previously-proposed methods to disentangle input-driven versus autonomous dynamics require repeatable trial structure [10, 19, 54]. Our work focuses only on the effects of unobserved neurons, and does not attempt to address this additional source of mechanistic mismatch. Observation of a network under

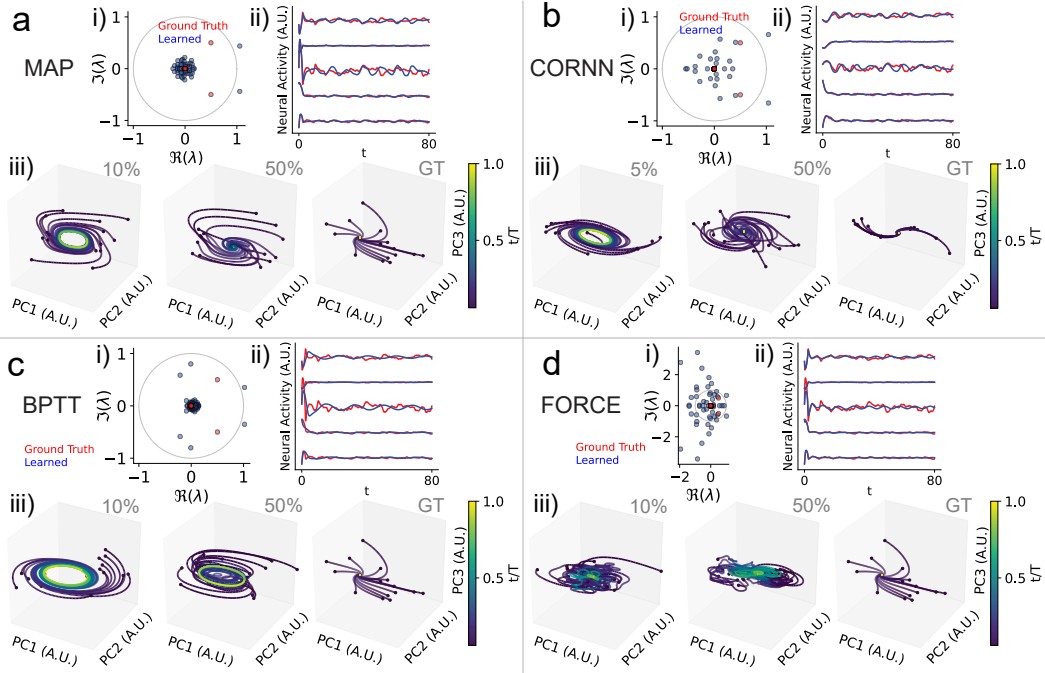

Figure 5: Eigenvalue overestimation leads to spurious limit cycle discovery across diverse inference methods. **a.** Learning from a noise-driven teacher with low-rank non-normal connectivity using MAP estimation. i). Ground truth teacher (red) and learned student (blue) dynamics matrix eigenvalues at 10% subsampling. Note the conjugate pair of learned eigenvalues with real part greater than 1. ii). Activity traces for the teacher (red) and student (blue) networks at 10% subsampling. iii). Example student network dynamics for 10% and 50% subsampling compared to the ground truth (GT). **b.** As in **a**, but for the "Convex Optimization of Recurrent Neural Networks (CORNN)" algorithm proposed by [14]. Since CORNN was proposed for leaky-rate (as opposed to leaky-current) dynamics, we modify the student and teacher dynamics accordingly. A spurious limit cycle is fit at 5% subsampling. **c.** As in **a**, but using backpropagation through time (BPTT). **d.** As in **a**, but for the recursive-least-squares based FORCE algorithm [51]. See Appendix G for detailed numerical methods.

a restricted set of input conditions poses a particularly striking challenge if it has a spectrum of heterogeneous integration timescales. If inputs drive activity along only a subset of dimensions, it is easy to imagine how a heterogeneous spectrum of time constants could be reductively interpreted as low-dimensional attractor structure [38, 39, 55]. When studying circuits whose upstream inputs are not well-understood dynamically, inferences about circuit-intrinsic attractor structure become even more tenuous.

As our work shows that data-constrained models can fail to correctly distinguish between mechanistically different hypotheses for the circuit basis of slow dynamics, an important question is how one should validate putative attractor structure. Though we do not address this issue in the present work, the obvious candidate for conclusive validation of attractor dynamics is of course direct experimental perturbation of neural activity. Daie, Svoboda, and colleagues have identified feedforward amplification (in lieu of line-attractor-like dynamics) in anterior lateral motor cortex using the correlation structure of responses to optogenetic perturbations [36]. O'Shea, Duncker, and colleagues have used targeted optogenetic perturbations to interrogate putative low-dimensional dynamics in primate motor cortex [25]. There, they show that data-constrained models with unconstrained weight matrices do not readily predict perturbation responses, while those with weight matrices constrained to be low-rank capture the fast recovery of the dynamics to a low-dimensional subspace. After the completion of our work, Vinograd et al. [56] have begun to interrogate putative line attractor dynamics in hypothalamus using similar perturbations. An important question for future work will be to determine how specifically targeted a patterned optogenetic perturbation must be in order to distinguish between the line attractor and functionally-feedforward integrator networks studied here. Another important

question is how to relate patterned stimuli *in silico* and *in vivo*, particularly for latent variable models where a possibly ill-posed inversion of the latent state to observed neuron mapping would be required.

Finally, we remark that our work relates to a broader question of inference for partially-observed dynamical systems. There is a substantial literature on performance guarantees for parameter estimation for linear systems, which largely focuses on predictive accuracy rather than qualitative features [44–47, 57]. There are also a host of methods which leverage delay embeddings; in addition to predictive accuracy, these methods prioritize accurately inferring the dimensionality and topological structure of an underlying system [20, 58–60]. However, such methods are "equation-free", and thus are not ideal for identifying how dynamical variables are coupled [61]. Consequently, such methods might not be suitable for distinguishing between mechanistically distinct circuit hypotheses that generate similarly low dimensional neural activity. Indeed, delay embeddings do not appear to distinguish between the two integrator circuits studied here (Fig. G.7). We leave a more detailed investigation of delay embedding-based approaches for future work.

## Acknowledgments and Disclosure of Funding

We thank Farhad Pashakhanloo and Mitchell Ostrow for helpful comments on a previous version of our manuscript. JAZV and CP were supported by NSF Award DMS-2134157 and NSF CAREER Award IIS-2239780. JAZV was further supported by a Junior Fellowship from the Harvard Society of Fellows. CP was further supported by a Sloan Research Fellowship. This work has been made possible in part by a gift from the Chan Zuckerberg Initiative Foundation to establish the Kempner Institute for the Study of Natural and Artificial Intelligence.

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

# A  Introduction to integrator models

In this appendix, we provide a brief, pedagogical introduction to the integrator models used as motivating examples in §2. We recall from the main text that both models have dynamics

$$\tau\dot{\mathbf{z}} = -\mathbf{z} + J\mathbf{z} + \mathbf{b}u$$

for state $\mathbf{z} \in \mathbb{R}^D$, recurrent weights $J \in \mathbb{R}^{D \times D}$, and input $u(t) \in \mathbb{R}$ encoded through a vector $\mathbf{b} \in \mathbb{R}^D$. They differ only in the choice of weight matrix $J$. These linear dynamics are of course exactly solvable, yielding

$$\mathbf{z}(t) = e^{(J-I_D)t/\tau}\mathbf{z}(0) + \int_0^t \frac{ds}{\tau}e^{(J-I_D)(t-s)/\tau}\mathbf{b}u(s).$$

## A.1  Line attractor

The construction of the classic line attractor network as popularized by Seung [28] starts by assuming that $J$ is symmetric, such that it admits an orthogonal eigendecomposition with real eigenvalues

$$J = O\Lambda O^\top$$

for $OO^\top = O^\top O = I_D$ and $\Lambda = \mathrm{diag}(\lambda_1, \ldots, \lambda_D)$ for $\lambda_j \in \mathbb{R}$. We assume that the eigenvalues are ordered as $\lambda_1 \geq \lambda_2 \geq \cdots \geq \lambda_D$. For the system to be stable, we must of course have $\lambda_j \leq 1$ for all $j$. Then, letting

$$\tilde{\mathbf{z}}(t) = O^\top \mathbf{z}(t)$$

and

$$\tilde{\mathbf{b}} = O^\top \mathbf{b}$$

be the projections of the state and encoding vector into the eigenvector basis, we have

$$\tilde{z}_j(t) = e^{-t/\tau_j}\tilde{z}_j(0) + \tilde{b}_j \int_0^t \frac{ds}{\tau}e^{-(t-s)/\tau_j}u(s),$$

where we have introduced the timescales

$$\tau_j = \frac{\tau}{1 - \lambda_j}.$$

Then, it is easy to see that if for some $j$ we have $\lambda_j = 1$, the corresponding timescale $\tau_j$ will be infinite and the activity $\tilde{z}_j(t)$ along that dimension will perfectly integrate $u(t)$. If integrating $u(t)$ in a way that is stable to perturbations of the network is our only goal, then activity along other dimensions should decay in time, meaning that we should have all other eigenvalues be strictly less than one, i.e., $1 = \lambda_1 > \lambda_2 \geq \cdots \geq \lambda_D$. Moreover, we should have $\tilde{b}_k = 0$ for all $k > 1$, i.e., the input should be aligned to the top eigenvector of $J$. For the decay to be fast, we want the gap between $\lambda_1$ and $\lambda_2$ to be large. The classic line attractor network achieves this very simply, choosing

$$J_{ij} = \begin{cases} 0 & i = j \\ 1/(D-1) & i \neq j \end{cases},$$

such that it has eigenvalue 1 with multiplicity 1, corresponding to an eigenvector proportional to $\mathbf{1}_D$, and eigenvalue $-1/(D-1)$ with multiplicity $D-1$ [28, 40].

However, in a realistic setting, it will not be possible to fine-tune the top eigenvalue exactly to 1, and there will be some decay along the integration dimension. Therefore, one must consider *approximate* line attractor dynamics, for which $\lambda_1 = 1 - \varepsilon$ for some error $\varepsilon > 0$, while the other eigenvalues are far smaller, i.e., $\lambda_1 \gg \lambda_2 \geq \cdots \geq \lambda_D$ [13, 27, 28, 40]. This network is exceptionally sensitive to the error $\varepsilon$, as with $\lambda_1 = 1 - \varepsilon$ one has $\tau_1 = \tau/\varepsilon$, and the error between the true integral of $u(t)$ and the readout from the approximate attractor network is exponentially large in time. Yet, so long as $\lambda_1 \gg \lambda_2$, perturbations along the approximate integration dimension will still decay exponentially more slowly than those along other dimensions.

In Figure 1, we generated connectivity $J$ such that the largest eigenvalue is close to 1, and all other eigenvalues are $< 1$. Specifically, we used $J = Q\Lambda Q^{-1}$ for

$$\Lambda_{ij} = \begin{cases} 1 - 10^{-3} & i = j = 1 \\ 0.2 & i = j \geq 2 \\ 0 & i \neq j \end{cases},$$

and $Q$ a matrix generated with entries $Q_{ij} \sim \mathcal{N}(0, \frac{1}{\sqrt{D}})$. Note that for realism, we have relaxed the symmetry constraint, and instead use connectivity that can be related to a corresponding symmetric approximate line attractor via a similarity transform. We use $D = 500$ as the size of the network.

### A.2 Functionally-feedforward integrator

The exquisite sensitivity of the line attractor network to small perturbations of the synaptic weights has motivated theoretical investigation of a panoply of alternative integrator circuits. Restricting our attention to simple linear networks, the most prominent proposal is approximate integration through functionally-feedforward non-normal integration [32, 37]. This model starts with the following linear-algebraic observation: if $J$ is non-normal (i.e., $JJ^\top \neq J^\top J$), though one loses orthogonal diagonalizability, one can still consider the Schur decomposition

$$J = OTO^\top,$$

where $O$ is orthogonal and $T$ is upper triangular. As proposed by Goldman [32], the Schur decomposition is a more conceptually useful tool for interpreting non-normal dynamics than the eigendecomposition, as it preserves the orthogonality of the modes. In particular, while if the dynamics are normal $T$ is diagonal and each mode only excites itself, if $J$ is non-normal a given mode may interact 'later' modes in a hidden feedforward structure, revealing a circuit basis for non-normal amplification.

As the simplest example of this structure, Goldman [32] considered a hidden chain structure

$$T_{ij} = \delta_{i+1,j}.$$

As $T$ is strictly upper triangular, all eigenvalues of $J$ vanish. Considering the mode decomposition

$$\tilde{\mathbf{z}}(t) = O^\top \mathbf{z}(t)$$

and

$$\tilde{\mathbf{b}} = O^\top \mathbf{b}$$

as we did in the symmetric case, we have the mode-wide dynamics

$$\tau \dot{\tilde{z}}_{j+1}(t) = -\tilde{z}_{j+1}(t) + \tilde{z}_j(t) + \tilde{b}_{j+1} u(t).$$

This gives sequential low-pass filtering of the input, which allows approximate maintenance of a memory over $\mathcal{O}(\tau D)$ time [32, 37]. Importantly, this mechanism is inherently far less sensitive to small variations in the weights than the line attractor.

For the functionally feedforward network in Figure 1, we use

$$T_{ij} = \delta_{i+1,j} + \beta \delta_{i,1}(1 - \delta_{1,j}).$$

Here, $\beta$ controls the strength of skip connections that further amplify the output mode of activity. We select $\beta = 0.5$ so that, like the line attractor network, the activity produced by the functionally feedforward network is approximately low-dimensional. We generate $O$ as an orthonormal matrix uniformly at random with respect to the Haar measure, and use $D = 500$ as the size of the network. For input weights, we use the sum of the Schur modes $\mathbf{b} = \sum_{i=1}^{D} O_{:,i}$, where $O_{:,i}$ denotes the $i$th Schur mode. Then, any readout proportional to the mean Schur mode will then solve the integration task up to a constant rescaling. To achieve the correct readout scale for $\beta = 0.5$, we used readout weights $0.7 \cdot \overline{O}$, where $\overline{O} = \frac{1}{D} \sum_{i=1}^{D} O_{:,i}$ denotes the mean Schur mode.

## B  MAP inference of connectivity in noise-driven RNNs

In this Appendix, we lay out the procedure sketched in §3 for maximum *a posteriori* (MAP) inference of connectivity in noise-driven RNNs that underlies our analytical results. We first consider the continuous-time setting directly, and then the discretized case.

## B.1 Continuous time

We first consider the continuous-time setting. We recall from the main text that we consider a teacher-student setup, where the teacher has $D$ neurons and a recurrent weight matrix $B$, such that the dynamics of its firing rate vector $\mathbf{z}(t) \in \mathbb{R}^D$ is

$$\tau\dot{\mathbf{z}} = -\mathbf{z} + B\phi(\mathbf{z}) + \boldsymbol{\xi}(t)$$

where $\boldsymbol{\xi}(t)$ is uncorrelated Gaussian noise with $\mathbb{E}[\boldsymbol{\xi}(t)] = \mathbf{0}$ and $\mathbb{E}[\boldsymbol{\xi}(t)\boldsymbol{\xi}(s)^\top] = 2\sigma_\xi^2 \delta(t-s)I_D$, and $\phi$ is a possibly nonlinear transfer function, which we take to act elementwise. Again, we assume a $d$-dimensional student with recurrent weights $A$, such that its rate $\mathbf{x}(t) \in \mathbb{R}^d$ evolves as

$$\tau\dot{\mathbf{x}} = -\mathbf{x} + A\phi(\mathbf{x}) + \boldsymbol{\eta}(t),$$

where $\boldsymbol{\eta}(t)$ is $d$-dimensional white noise with $\mathbb{E}[\boldsymbol{\eta}(t)] = \mathbf{0}$ and $\mathbb{E}[\boldsymbol{\eta}(t)\boldsymbol{\eta}(s)^\top] = 2\sigma_\eta^2 \delta(t-s)I_d$. Assuming $d < D$, we observe the first $d$ neurons of the teacher:

$$\mathbf{x}^{\text{obs}}(t) = P\mathbf{z}(t) \quad \text{for} \quad t \in [0, T] \quad \text{and} \quad P = (I_d, \quad 0_{d\times(D-d)}).$$

Our goal is to infer the student's weight matrix $A$ given these observations.

To do so, we use MAP inference. Our starting point is the likelihood of observing a trajectory $\{\mathbf{x}^{\text{obs}}(t) : t \in [0, T]\}$ given a particular weight matrix $A$, which using the path integral representation of an Itô process can be written non-rigorously as

$$p(\{\mathbf{x}^{\text{obs}}(t) : t \in [0, T]\} \,|\, A) \propto \exp\left[-\frac{1}{2\sigma_\eta^2}\int_0^T dt\, \|\tau\dot{\mathbf{x}}^{\text{obs}}(t) + \mathbf{x}^{\text{obs}}(t) - A\phi(\mathbf{x}^{\text{obs}}(t))\|^2\right].$$

Here, we have used that

$$\phi(\mathbf{x}^{\text{obs}}) = \phi(P\mathbf{z}) = P\phi(\mathbf{z})$$

to simplify the notation. To make the problem analytically tractable, we choose an isotropic Gaussian prior over the elements of $A$:

$$A_{ij} \sim_{\text{i.i.d.}} \mathcal{N}\left(0, \frac{\sigma_\eta^2}{T\rho}\right)$$

where $\rho > 0$. We have chosen this parameterization of the prior variance because it makes the log-posterior density particularly simple:

$$L = -\frac{\sigma_\eta^2}{T}\log p(A \,|\, \{\mathbf{x}^{\text{obs}}(t) : t \in [0, T]\})$$

$$= \int_0^T \frac{dt}{T}\|\tau\dot{\mathbf{x}}^{\text{obs}}(t) + \mathbf{x}^{\text{obs}}(t) - A\phi(\mathbf{x}^{\text{obs}}(t))\|^2 + \rho\|A\|_F^2.$$

We remark that we have proceeded rather cavalierly in our treatment of the functional density, but this procedure can equally well be viewed as ridge-regularized least-squares estimation. We will also arrive at the same characterization of the log-posterior density as the continuous-time limit of the discrete setting in the subsequent subsection.

As the log-posterior density is quadratic, it is easy to read off that the MAP estimate of $A$ is

$$\hat{A}_T = \left[\int_0^T \frac{dt}{T}[\tau\dot{\mathbf{x}}^{\text{obs}}(t) + \mathbf{x}^{\text{obs}}(t)]\phi(\mathbf{x}^{\text{obs}}(t))^\top\right]\left[\int_0^T \frac{dt}{T}\phi(\mathbf{x}^{\text{obs}}(t))\phi(\mathbf{x}^{\text{obs}}(t))^\top + \rho I_d\right]^{-1},$$

where we add a subscript $T$ to emphasize the observation window. Using the dynamics of $\mathbf{x}^{\text{obs}}(t) = P\mathbf{z}(t)$, we can re-write this in terms of the teacher's dynamics as

$$\hat{A}_T = P\left[BC_T + \int_0^T \frac{dt}{T}\boldsymbol{\xi}(t)\phi(\mathbf{z}(t))^\top\right]P^\top\left[PC_TP^\top + \rho I_d\right]^{-1} \tag{B.1}$$

where

$$C_T = \int_0^T \frac{dt}{T}\phi(\mathbf{z}(t))\phi(\mathbf{z}(t))^\top$$

is the empirical covariance of the teacher network activity.

So far, we have let $\phi$ be general. However, we now specialize to the linear setting $\phi(z) = z$, in which the student and the teacher are Ornstein–Uhlenbeck (OU) processes. Then, we have the formal solution

$$\mathbf{z}(t) = e^{(-I_D + B)t}\mathbf{z}(0) + \int_0^t ds\, e^{(-I_D + B)(t-s)}\boldsymbol{\xi}(s),$$

and, at least in the long-time limit, we can leverage the classical theory of such processes [43, 48].

Provided that all eigenvalues of the dynamics matrix $-I_D + B$ have negative real part, this process will converge to a Gaussian stationary state with equal-time covariance

$$\mathbb{E}_s[\mathbf{z}(t)\mathbf{z}(t)^\top] = S$$

which solves the Lyapunov equation

$$(I_D - B)S + S(I_D - B) = 2\sigma_\xi^2 I_D,$$

or equivalently is given by the matrix integral

$$S = 2\sigma_\xi^2 \int_0^\infty dt\, e^{-(I_D - B)t}e^{-(I_D - B)^\top t}.$$

In the stationary state, the time-lagged correlation

$$C(\tau) = \mathbb{E}_s[\mathbf{z}(t)\mathbf{z}(t + \tau)^\top]$$

is given by

$$C(\tau) = e^{-(I_D - B)\tau}S$$

Moreover, if one adds infinitesimal linear perturbations to the dynamics as

$$\dot{\mathbf{z}}(t) = (-I_D + B)\mathbf{z} + \boldsymbol{\eta}(t) + \mathbf{h}(t),$$

one has that the linear response to perturbations of the system in the stationary state is given by

$$R_{ij}(\tau) = \frac{\delta \mathbb{E}_s[z_i(t + \tau)]}{\delta h_j(t)} = e^{-(I_D - B)\tau}$$

so that

$$C(\tau) = R(\tau)S.$$

Thus, we will have

$$\lim_{T \to \infty} C_T = S,$$

and we claim that

$$\lim_{T \to \infty} \int_0^T \frac{dt}{T} \boldsymbol{\xi}(t)\mathbf{z}(t)^\top = 0.$$

The vanishing of this term follows from the observation that

$$\mathbb{E}\left[\int_0^T \frac{dt}{T} \boldsymbol{\xi}(t)\mathbf{z}(t)^\top\right] = \mathbf{0}$$

while by the Itô isometry

$$\mathbb{E}\left[\left(\int_0^T \frac{dt}{T} \boldsymbol{\xi}(t)\mathbf{z}(t)^\top\right)_{ij}\left(\int_0^T \frac{dt}{T} \boldsymbol{\xi}(t)\mathbf{z}(t)^\top\right)_{i'j'}\right] = \frac{1}{T}\delta_{ii'}\sigma_\xi^2 \int_0^T \frac{dt}{T}\mathbb{E}[z_j(t)z_{j'}(t)].$$

Thus, from (B.1), we conclude that the MAP estimated student dynamics matrix in the long time limit takes the form

$$\hat{A}_\infty = PBSP^\top \left(PSP^\top + \rho I_d\right)^{-1}. \tag{B.2}$$

As an aside, if $B$ is a symmetric matrix, the process will be reversible, and the stationary state an equilbrium. In this case, setting $\sigma_\xi^2 = 1$ for brevity, the stationary covariance takes the relatively simple form

$$S = \int_0^\infty dt\, e^{-(I_D - B)t} = (I_D - B)^{-1}. \tag{B.3}$$

In this case, we can gain some intuition for the effect of partial observation directly from considering the stationary covariance. Consider a generic symmetric weight matrix, partitioned according to the

observed and non-observed neurons:

$$B = \begin{pmatrix} B_{oo} & B_{on} \\ B_{on}^\top & B_{nn} \end{pmatrix}$$

We can then write the marginal covariance matrix of the observed neurons as

$$S_{oo} = [I_d - B_{oo} - B_{on}(I_{D-d} - B_{nn})^{-1}B_{on}^\top]^{-1}.$$

We can then interpret

$$B_{oo} + B_{on}(I_{D-d} - B_{nn})^{-1}B_{on}^\top$$

as a sort of effective weight matrix that accounts for the effect of feedback through the unobserved neurons on the stationary state.

## B.2 Discrete time

We now consider the discrete-time setting, in which the teacher and student are both $\mathrm{AR}(1)$ processes. Our goal here is to show that taking the continuum limit of the resulting estimate of the dynamics matrix recovers the result obtained directly in continuous time. Letting

$$\alpha = \frac{\Delta t}{\tau}$$

be the discretization scale, the teacher's dynamics are now

$$\mathbf{z}_t = (1 - \alpha)\mathbf{z}_{t-1} + \alpha B\phi(\mathbf{z}_{t-1}) + \sqrt{2\alpha}\boldsymbol{\xi}_t, \tag{B.4}$$

where $\boldsymbol{\xi}_t \sim \mathcal{N}(\mathbf{0}, \sigma_\xi^2 I_D)$ is isotropic Gaussian noise, while those of the student are

$$\mathbf{x}_t = (1 - \alpha)\mathbf{x}_{t-1} + \alpha A\phi(\mathbf{x}_{t-1}) + \sqrt{2\alpha}\boldsymbol{\eta}_t.$$

where $\boldsymbol{\eta}_t \sim \mathcal{N}(\mathbf{0}, \sigma_\eta^2 I_d)$ is isotropic Gaussian noise. The likelihood of some observed data $\{\mathbf{x}_t^{\mathrm{obs}}\}_{t \in [T]}$ is then given by

$$P(\{\mathbf{x}_t^{\mathrm{obs}}\}_{t \in [T]}|A) \propto \prod_{t=1}^T \exp\left(-\frac{1}{2\sigma_\eta^2 \alpha}||(1 - \alpha)\mathbf{x}_{t-1}^{\mathrm{obs}} + \alpha A\phi(\mathbf{x}_{t-1}^{\mathrm{obs}}) - \mathbf{x}_t^{\mathrm{obs}}||_2^2\right),$$

which is precisely the time-sliced analogue of the functional density considered above. Again assuming an isotropic Gaussian prior on the entries of $A$, we obtain the corresponding loss function

$$L = \frac{1}{T}\sum_{t=1}^T ||(1 - \alpha)\mathbf{x}_{t-1}^o + \alpha A\phi(\mathbf{x}_{t-1}^o) - \mathbf{x}_t^o||_2^2 + \rho||A||_2^2$$

where $\rho$ corresponds to the strength of the prior/regularization. We then can arrive at the MAP estimate of the dynamics matrix

$$\hat{A}_T = \alpha\left(\frac{1}{T}\sum_{t=1}^T (\mathbf{x}_t^o - (1 - \alpha)\mathbf{x}_{t-1}^o)\phi(\mathbf{x}_{t-1}^o)^\top\right)\left(\rho I + \alpha^2 \frac{1}{T}\sum_{t=1}^T \phi(\mathbf{x}_{t-1}^o)\phi(\mathbf{x}_{t-1}^o)^\top\right)^{-1}.$$

Again assuming that the observed data $\{\mathbf{x}_t^{\mathrm{obs}}\}_{t \in [T]}$ are produced via partial observations of the teacher activity

$$\mathbf{x}_t^{\mathrm{obs}} = P\mathbf{z}_t, \quad P = \begin{pmatrix} I_{d\times d} & \mathbf{0}_{d\times(D-d)} \end{pmatrix},$$

we can then describe the learned dynamics matrix solely in terms of properties of the teacher RNN:

$$\hat{A}_T = \alpha^2 P\left(BC_T + \frac{1}{T}\sum_{t=1}^T \boldsymbol{\xi}_t\phi(\mathbf{z}_{t-1})^\top\right)P^\top\left(\rho I_d + \alpha^2 PC_T P^\top\right)^{-1},$$

where

$$C_T = \frac{1}{T}\sum_{t=1}^T \phi(\mathbf{z}_{t-1})\phi(\mathbf{z}_{t-1})^\top.$$

It is now easy to see that the continuum limit of this discrete-time estimate converges in distribution to the continuous-time result.

In discrete time, it is easy to see that

$$\mathbb{E}\left[\frac{1}{T}\sum_{t=1}^{T}\boldsymbol{\xi}_t\phi(\mathbf{z}_{t-1})^\top\right] = \mathbf{0}$$

and

$$\mathbb{E}\left[\left(\frac{1}{T}\sum_{t=1}^{T}\boldsymbol{\xi}_t\phi(\mathbf{z}_{t-1})^\top\right)_{ij}\left(\frac{1}{T}\sum_{t=1}^{T}\boldsymbol{\xi}_t\phi(\mathbf{z}_{t-1})^\top\right)_{i'j'}\right]$$

$$= \frac{1}{T^2}\sum_{t=1}^{T}\mathbb{E}[\xi_{t,i}\phi(z_{t-1,j})\xi_{t,i'}\phi(z_{t-1,j'})] + \frac{1}{T^2}\sum_{t=1}^{T}\sum_{s\neq t}\mathbb{E}[\xi_{t,i}\phi(z_{t-1,j})\xi_{s,i'}\phi(z_{s-1,j'})]$$

$$= \delta_{ii'}\sigma_\xi^2\frac{1}{T^2}\sum_{t=1}^{T}\mathbb{E}[\phi(z_{t-1,j})\phi(z_{t-1,j'})]$$

$$+ \frac{1}{T^2}\sum_{t=1}^{T}\sum_{s>t}\mathbb{E}[\xi_{t,i}\phi(z_{t-1,j})\phi(z_{s-1,j'})]\mathbb{E}[\xi_{s,i'}]$$

$$+ \frac{1}{T^2}\sum_{t=1}^{T}\sum_{s<t}\mathbb{E}[\phi(z_{t-1,j})\xi_{s,i'}\phi(z_{s-1,j'})]\mathbb{E}[\xi_{t,i}]$$

$$= \delta_{ii'}\sigma_\xi^2\frac{1}{T^2}\sum_{t=1}^{T}\mathbb{E}[\phi(z_{t-1,j})\phi(z_{t-1,j'})]$$

$$= \frac{1}{T}\delta_{ii'}\sigma_\xi^2(C_T)_{jj'},$$

as $\mathbf{z}_{t-1}$ is independent of $\boldsymbol{\xi}_t$. Then, so long as $C_T$ remains bounded, this correlator tends in probability to zero as $T\to\infty$.

We thus arrive at the MAP estimate of the student dynamics matrix in the long time limit:

$$\hat{A}_\infty = \alpha^2 PBC_\infty P^\top\left(\rho I_d + \alpha^2 PC_\infty P^\top\right)^{-1},$$

the discrete time analog of (B.2). If we specialize to the linear case, letting

$$J = (1-\alpha)I_D + \alpha B$$

such that

$$\mathbf{z}_t = J\mathbf{z}_{t-1} + \sqrt{2\alpha}\boldsymbol{\xi}_t,$$

we have the formal solution

$$\mathbf{z}_t = J^t\mathbf{z}_0 + \sqrt{2\alpha}\sum_{k=1}^{t}J^{t-k}\boldsymbol{\xi}_k.$$

## B.3   A note on time constants

We note an equivalence between the time constants

$$\tau_i = \frac{\tau}{|1-\Re\lambda_i|}$$

used in this work and the discrete time analog used in previous work [13, 27, 30],

$$\tau_i' = \left|\frac{1}{\ln|\lambda_i'|}\right|,$$

where $\lambda_i'$ are the eigenvalues of the discrete-time dynamics matrix $J = (1-\alpha)I_D + \alpha B$, which in terms of the eigenvalues $\lambda_i$ of $B$ has eigenvalues $\lambda_i' = 1 - \alpha + \alpha\lambda_i$. Thus, $|\lambda_i'| = \sqrt{(1-\alpha+\alpha\Re\lambda_i)^2 + (\alpha\Im\lambda_i)^2}$. Taylor-expanding the logarithm yields

$$(\Delta t)\tau_i' = \left|\frac{\tau}{(1-\Re\lambda_i) + \mathcal{O}(\alpha)}\right|$$

or, in the true continuous-time limit,

$$\lim_{\Delta t \downarrow 0} (\Delta t)\tau_i' = \frac{\tau}{|1 - \Re\lambda_i|},$$

which matches the continuous-time time constants. For $\alpha \ll 1 - \Re\lambda_i$, we therefore may use the continuous-time result with negligible error.

## C   Normal teacher

In this Appendix, we derive the two results on normal teachers claimed in §3.1 of the main text: that the student eigenvalues are contained within the support of the teacher spectrum, and that an approximate line attractor is recovered by MAP inference even under severe partial observation.

Assuming that $BB^\top = B^\top B$, for large $T$, we can simplify the teacher covariance as follows:

$$C_T = \frac{1}{T}\sum_{t=1}^{T} \mathbf{z}_{t-1}\mathbf{z}_{t-1}^\top$$

$$\approx \frac{1}{T}\sum_{t=1}^{T} J^{t-1}\mathbf{z}_0\mathbf{z}_0^\top(J^\top)^{t-1} + \frac{2\alpha\sigma_\xi^2}{T}\sum_{t=2}^{T}\sum_{\tau=1}^{t-1} J^{t-1-\tau}(J^\top)^{t-1-\tau},$$

as only the diagonal noise terms should contribute. Further, the first sum above should also tend to $\mathbf{0}$ assuming that $J$ only has eigenvalues with modulus $< 1$. Note that if $\alpha$ is small, this condition is equivalent to the real parts of eigenvalues of $B$ being $< 1$. Simplifying the remaining term, we have

$$\frac{2\alpha\sigma_\xi^2}{T}\sum_{t=2}^{T}\sum_{\tau=1}^{t-1} J^{t-1-\tau}(J^\top)^{t-1-\tau} = \frac{2\alpha\sigma_\xi^2}{T}\sum_{t=2}^{T}\sum_{\tau=1}^{t-1}(JJ^\top)^{t-1-\tau}$$

$$= \frac{2\alpha\sigma_\xi^2}{T}\sum_{t=2}^{T}(JJ^\top)^{t-1}\sum_{\tau=1}^{t-1}(JJ^\top)^{-\tau},$$

where the first equality above follows from normality of $B$. Summing this Neumann series, this simplifies to

$$= \frac{2\alpha\sigma_\xi^2}{T}\sum_{t=2}^{T}(JJ^\top)^{t-1}((I_D - (JJ^\top)^{-1})^{-1}(I_D - (JJ^\top)^{-t}) - I_D)$$

$$= \frac{2\alpha\sigma_\xi^2}{T}\left[\left((I_D - (JJ^\top)^{-1})^{-1} - I_D)\sum_{t=2}^{T}(JJ^\top)^{t-1}\right) - (I_D - (JJ^\top)^{-1})^{-1}(JJ^\top)^{-1}(T - 2)\right].$$

The only term that remains in the limit $T \to \infty$ is

$$-\frac{2\alpha\sigma_\xi^2}{T}(I_D - (JJ^\top)^{-1})^{-1}(JJ^\top)^{-1}T = 2\alpha\sigma_\xi^2(I_D - JJ^\top)^{-1},$$

yielding

$$C_\infty = 2\alpha\sigma_\xi^2(I_D - JJ^\top)^{-1}.$$

Expanding,

$$JJ^\top = (1-\alpha)^2 I_D + (1-\alpha)\alpha(B + B^\top) + \alpha^2 BB^\top$$

$$= I_D + 2\alpha(B_s - I_D) + \alpha^2(BB^\top - 2B_s),$$

where we have defined $B_s = \frac{B+B^\top}{2}$ as the symmetric part of $B$. This yields the simplification

$$C_\infty = \sigma_\xi^2\left[I_D - B_s + \alpha\left(B_s - \frac{1}{2}BB^\top\right)\right]^{-1}.$$

For $\alpha \ll 1$, we recover the continuous time result for symmetric $B$ stated in (B.3).

Thus, we arrive at the expression

$$\hat{A}_\infty = PB(I_D - B_s)^{-1}P^\top(\tilde{\rho}I_d + P(I_D - B_s)^{-1}P^\top)^{-1}$$

where the regularization $\tilde{\rho}$ has been re-scaled appropriately to absorb constants.

Since $B$ is normal, it can be diagonalized over $\mathbb{C}$ as $B = U\Lambda U^*$. Observe that $\frac{B+B^\top}{2} = \frac{B+B^*}{2} = U\Re(\Lambda)U^*$ where $\Re$ denotes the real part. We then have that

$$\hat{A}_\infty = PU\Lambda(I_D - \Re(\Lambda))^{-1}U^*P^\top \left(\tilde{\rho}I_d + PU(I_D - \Re(\Lambda))^{-1}U^*P^\top\right)^{-1}$$

$$= \left(\sum_{i=1}^{D} \frac{\lambda_i}{1 - \Re(\lambda_i)}\mathbf{u}_{1:d}^i(\mathbf{u}_{1:d}^i)^*\right)\left(\tilde{\rho}I_d + \sum_{i=1}^{D} \frac{1}{1 - \Re(\lambda_i)}\mathbf{u}_{1:d}^i(\mathbf{u}_{1:d}^i)^*\right)^{-1} \quad \text{(C.1)}$$

where $\mathbf{u}_{1:d}^i$ represents the truncated/projected $i$th eigenvector of $B$. For brevity, define $K = \left(\tilde{\rho}I_d + \sum_{i=1}^{D} \frac{1}{1 - \Re(\lambda_i)}\mathbf{u}_{1:d}^i(\mathbf{u}_{1:d}^i)^*\right)$.

We first analyze the symmetric case $B = B^\top$, where the argument is slightly simpler. The more general normal case is addressed in C.1. In this case, we have:

$$\hat{A}_\infty = \left(\sum_{i=1}^{D} \frac{\lambda_i}{1 - \lambda_i}\mathbf{u}_{1:d}^i(\mathbf{u}_{1:d}^i)^\top\right)K^{-1} \quad \text{(C.2)}$$

Using the fact that for general matrices $P$ and $Q$, $PQ$ and $QP$ have the same eigenvalues, we can instead analyze the spectrum of

$$\hat{A}'_\infty = K^{-1/2}\left(\sum_{i=1}^{D} \frac{\lambda_i}{1 - \lambda_i}\mathbf{u}_{1:d}^i(\mathbf{u}_{1:d}^i)^\top\right)K^{-1/2}.$$

Using that $\lambda_1 \geq \lambda_i$ for all $i$, we have

$$\hat{A}'_\infty \preceq K^{-1/2}\left(\sum_{i=1}^{D} \frac{\lambda_1}{1 - \lambda_i}\mathbf{u}_{1:d}^i(\mathbf{u}_{1:d}^i)^\top\right)K^{-1/2}$$

where $P \preceq Q$ denotes that $P - Q$ is negative semidefinite.

If we further suppose that $\tilde{\rho}\lambda_1 \geq 0$, we have the relation

$$K^{-1/2}\left(\sum_{i=1}^{D} \frac{\lambda_1}{1 - \lambda_i}\mathbf{u}_{1:d}^i(\mathbf{u}_{1:d}^i)^\top\right)K^{-1/2} \preceq K^{-1/2}\left(\tilde{\rho}\lambda_1 I_d + \sum_{i=1}^{D} \frac{\lambda_1}{1 - \lambda_i}\mathbf{u}_{1:d}^i(\mathbf{u}_{1:d}^i)^\top\right)K^{-1/2}$$

$$= K^{-1/2}(\lambda_1 K)K^{-1/2}$$

$$= \lambda_1 I_d$$

Thus, if $\tilde{\rho}\lambda_1 \geq 0$, all eigenvalues of $\hat{A}_\infty$ satisfy $\hat{\lambda}_i \leq \lambda_1$. Similarly, if $\tilde{\rho}\lambda_D \leq 0$, all eigenvalues of $\hat{A}_\infty$ satisfy $\hat{\lambda}_i \geq \lambda_D$. Both upper and lower bounds are necessarily satisfied simultaneously in the ridgeless limit $\rho \to 0$.

## C.1 More general normal matrix case

Suppose $B$ has $p$ pairs of complex eigenvalues $\lambda_{c_j}, \overline{\lambda}_{c_j}$ with corresponding eigenvectors $\mathbf{u}^{c_j}, \overline{\mathbf{u}}^{c_j}$, as well as $D - 2p$ real eigenvalues $\lambda_{r_j}$ with corresponding eigenvectors $\mathbf{u}^{r_j}$. We can then rewrite (C.1) as

$$\hat{A}_\infty = \left(2\sum_{j=1}^{p} \frac{\Re(\lambda_{c_j})}{1 - \Re(\lambda_{c_j})}F_{c_j} + 2\sum_{j=1}^{p} \frac{\Im(\lambda_{c_j})}{1 - \Re(\lambda_{c_j})}G_{c_j} + \sum_{j=1}^{D-2p} \mathbf{u}_{1:d}^{r_j}(\mathbf{u}_{1:d}^{r_j})^\top \frac{\lambda_{r_j}}{1 - \lambda_{r_j}}\right)K^{-1}$$

where

$$F_{c_j} = \left(\Re(\mathbf{u}_{1:d}^{c_j})\Re(\mathbf{u}_{1:d}^{c_j})^\top + \Im(\mathbf{u}_{1:d}^{c_j})\Im(\mathbf{u}_{1:d}^{c_j})^\top\right)$$

and

$$G_{c_j} = \left(\Re(\mathbf{u}_{1:d}^{c_j})\Im(\mathbf{u}_{1:d}^{c_j})^\top - \Im(\mathbf{u}_{1:d}^{c_j})\Re(\mathbf{u}_{1:d}^{c_j})^\top\right)$$

. Similarly, we can also express $K$ in terms of real components:

$$K = \tilde{\rho}I_d + 2\sum_{j=1}^{p} \frac{1}{1 - \Re(\lambda_{c_j})}F_{c_j} + \sum_{j=1}^{D-2p} \frac{1}{1 - \lambda_{r_j}}\mathbf{u}_{1:d}^{r_j}(\mathbf{u}_{1:d}^{r_j})^\top.$$

We then study the spectrum of

$$\hat{A}'_\infty = K^{-1/2} \left( 2 \sum_{j=1}^{p} \frac{\Re(\lambda_{c_j})}{1 - \Re(\lambda_{c_j})} F_{c_j} + 2 \sum_{j=1}^{p} \frac{\Im(\lambda_{c_j})}{1 - \Re(\lambda_{c_j})} G_{c_j} + \sum_{j=1}^{D-2p} \mathbf{u}_{1:d}^{r_j} \frac{\lambda_{r_j}}{1 - \lambda_{r_j}} (\mathbf{u}_{1:d}^{r_j})^\top \right) K^{-1/2}$$

$\hat{A}'_\infty$ is no longer symmetric because of the skew-symmetric components $G_{c_j}$. However, we can still analyze the symmetric component

$$(\hat{A}'_\infty)_s = K^{-1/2} \left( 2 \sum_{j=1}^{p} \frac{\Re(\lambda_{c_j})}{1 - \Re(\lambda_{c_j})} F_{c_j} + \sum_{j=1}^{D-2p} \mathbf{u}_{1:d}^{r_j} \frac{\lambda_{r_j}}{1 - \lambda_{r_j}} (\mathbf{u}_{1:d}^{r_j})^\top \right) K^{-1/2}.$$

As before, under the condition $\tilde{\rho}\Re(\lambda_1) \geq 0$, we have the ordering

$$(\hat{A}'_\infty)_s \preceq K^{-1/2} \left( \tilde{\rho}\Re(\lambda_1) I_d + 2 \sum_{j=1}^{p} \frac{\Re(\lambda_1)}{1 - \Re(\lambda_{c_j})} F_{c_j} + \sum_{j=1}^{D-2p} \mathbf{u}_{1:d}^{r_j} \frac{\Re(\lambda_1)}{1 - \lambda_{r_j}} (\mathbf{u}_{1:d}^{r_j})^\top \right) K^{-1/2}$$

$$= K^{-1/2}(\Re(\lambda_1)K)K^{-1/2}$$

$$= \Re(\lambda_1)I_d$$

We can then observe that $\mathbf{v}^\top (\hat{A}'_\infty - \Re(\lambda_1)I_d)\mathbf{v} \leq 0$ for arbitrary $\mathbf{v} \in \mathbb{R}^d$, since the symmetric component of $\hat{A}'_\infty - \Re(\lambda_1)I_d$ is NSD. This implies that $\Re(\hat{\lambda}_j) \leq \Re(\lambda_1)$ for all $j$.

Similarly, under the condition $\tilde{\rho}\lambda_D \leq 0$, $\Re(\hat{\lambda}_j) \geq \Re(\lambda_D)$ for all $j$. Both upper and lower bounds again hold simultaneously for $\rho \to 0$, regardless of the teacher spectra.

## C.2 Stronger result for symmetric teachers

In the symmetric case $B = B^\top$ with $\rho \to 0$, we can also show a stronger result that $\hat{\lambda}_j \leq \lambda_j$ for all $j \in \{1, \ldots d\}$:

Observe that for any $j > 1$, that

$$\hat{A}'_\infty = K^{-1/2} \left( \sum_{i=1}^{j-1} \frac{\lambda_i}{1 - \lambda_i} \mathbf{u}_{1:d}^i \mathbf{u}_{1:d}^i{}^\top \right) K^{-1/2} + K^{-1/2} \left( \sum_{i=j}^{D} \frac{\lambda_i}{1 - \lambda_i} \mathbf{u}_{1:d}^i \mathbf{u}_{1:d}^i{}^\top \right) K^{-1/2}$$

Let $\overline{\lambda}_j(\cdot)$ denote the $j$th largest eigenvalue of $\cdot$. Applying Weyl's inequality, we have

$$\overline{\lambda}_j(\hat{A}'_\infty) \leq \overline{\lambda}_j \left( K^{-1/2} \left( \sum_{i=1}^{j-1} \frac{\lambda_i}{1 - \lambda_i} \mathbf{u}_{1:d}^i \mathbf{u}_{1:d}^i{}^\top \right) K^{-1/2} \right)$$

$$+ \overline{\lambda}_1 \left( K^{-1/2} \left( \sum_{i=j}^{D} \frac{\lambda_i}{1 - \lambda_i} \mathbf{u}_{1:d}^i \mathbf{u}_{1:d}^i{}^\top \right) K^{-1/2} \right)$$

$$= \overline{\lambda}_1 \left( K^{-1/2} \left( \sum_{i=j}^{D} \frac{\lambda_i}{1 - \lambda_i} \mathbf{u}_{1:d}^i \mathbf{u}_{1:d}^i{}^\top \right) K^{-1/2} \right) \tag{C.3}$$

which follows since the first term of the RHS is of rank $\leq j - 1$.

We also have that

$$K^{-1/2} \left( \sum_{i=j}^{D} \frac{\lambda_i}{1 - \lambda_i} \mathbf{u}_{1:d}^i \mathbf{u}_{1:d}^i{}^\top \right) K^{-1/2} \preceq K^{-1/2} \left( \sum_{i=j}^{D} \frac{\lambda_j}{1 - \lambda_i} \mathbf{u}_{1:d}^i \mathbf{u}_{1:d}^i{}^\top \right) K^{-1/2}$$

$$= \lambda_j I_D$$

Thus, we can bound (C.3) above by $\lambda_j$, yielding the result

$$\hat{\lambda}_j \leq \lambda_j \tag{C.4}$$

## C.3 Line attractor recovery

Suppose the teacher is a near perfect symmetric line attractor. In particular, let $B = B^\top$ have eigenvalues $\lambda_1 = 1 - \varepsilon$, $\varepsilon \ll 1$, and $\lambda_i \ll 1$ for $i \geq 2$. For simplicity, assume $\rho \to 0$. In this case, we can express (C.2) as

$$\hat{A}_\infty = \left( \frac{1-\varepsilon}{\varepsilon} \mathbf{u}_{1:d}^1 (\mathbf{u}_{1:d}^1)^\top + \sum_{i=2}^D \frac{\lambda_i}{1-\lambda_i} \mathbf{u}_{1:d}^i (\mathbf{u}_{1:d}^i)^\top \right) \left( \frac{1}{\varepsilon} \mathbf{u}_{1:d}^1 (\mathbf{u}_{1:d}^1)^\top + \sum_{i=2}^D \frac{1}{1-\lambda_i} \mathbf{u}_{1:d}^i (\mathbf{u}_{1:d}^i)^\top \right)^{-1}$$
(C.5)

Denote $P_1 = \sum_{i=2}^D \frac{\lambda_i}{1-\lambda_i} \mathbf{u}_{1:d}^i (\mathbf{u}_{1:d}^i)^\top$ and $P_2 = \sum_{i=2}^D \frac{1}{1-\lambda_i} \mathbf{u}_{1:d}^i (\mathbf{u}_{1:d}^i)^\top$. From Weyl's perturbation bounds on symmetric matrices [62], we can bound the eigenvalues of the "numerator" as follows:

$$\left| \overline{\lambda}_1 \left( \frac{1-\varepsilon}{\varepsilon} \mathbf{u}_{1:d}^1 (\mathbf{u}_{1:d}^1)^\top + P_1 \right) - \overline{\lambda}_1 \left( \frac{1-\varepsilon}{\varepsilon} \mathbf{u}_{1:d}^1 (\mathbf{u}_{1:d}^1)^\top \right) \right| \leq ||P_1||_{\text{op}}$$

$$\leq \frac{\lambda_2}{1-\lambda_2} \qquad \text{(C.6)}$$

where (C.6) follows from the Cauchy interlacing theorem [63]. This yields the bound on the top eigenvalue of the numerator,

$$\overline{\lambda}_1 \left( \frac{1-\varepsilon}{\varepsilon} \mathbf{u}_{1:d}^1 (\mathbf{u}_{1:d}^1)^\top + P_1 \right) \geq \frac{1-\varepsilon}{\varepsilon} ||\mathbf{u}_{1:d}^1||_2^2 - \frac{\lambda_2}{1-\lambda_2}$$

We can obtain a similar bound on the largest eigenvalue of the "denominator":

$$\overline{\lambda}_1 \left( \frac{1}{\varepsilon} \mathbf{u}_{1:d}^1 (\mathbf{u}_{1:d}^1)^\top + P_2 \right) \leq \frac{1}{\varepsilon} ||\mathbf{u}_{1:d}^1||_2^2 + \frac{1}{1-\lambda_2}$$

In the case where $\lambda_i \geq 0$ (e.g., no timescale is faster than the intrinsic timescale of a single neuron), we can use bounds on the eigenvalues of products of PSD matrices to obtain the following:

$$
\begin{aligned}
\hat{\lambda}_1 = \overline{\lambda}_1(\hat{A}_\infty) &\geq \overline{\lambda}_1(\text{Num}) \overline{\lambda}_d(\text{Den}^{-1}) \\
&= \overline{\lambda}_1(\text{Num}) (\overline{\lambda}_1(\text{Den}))^{-1} \\
&\geq \left( \frac{1-\varepsilon}{\varepsilon} ||\mathbf{u}_{1:d}^1||_2^2 - \frac{\lambda_2}{1-\lambda_2} \right) \left( \frac{1}{\varepsilon} ||\mathbf{u}_{1:d}^1||_2^2 + \frac{1}{1-\lambda_2} \right)^{-1} \\
&\geq \left( \frac{1-\varepsilon}{\varepsilon} ||\mathbf{u}_{1:d}^1||_2^2 - \frac{1+\lambda_2}{1-\lambda_2} \right) \left( \frac{1}{\varepsilon} ||\mathbf{u}_{1:d}^1||_2^2 \right)^{-1} \\
&= \lambda_1 - \frac{\varepsilon(1+\lambda_2)}{||\mathbf{u}_{1:d}^1||_2^2 (1-\lambda_2)}
\end{aligned}
$$

where we have used 'Num' and 'Den' as shorthand for the factors in (C.5). Assuming eigendirections are randomly oriented, $||\mathbf{u}_{1:d}^1||_2^2 = \mathcal{O}\left( \frac{d}{D} \right)$.

From result (C.4), we have an upper bound on the second largest eigenvalue

$$\hat{\lambda}_2 \leq \lambda_2$$

Thus, under the stated assumptions, we can conclude $\hat{\lambda}_1 \geq \lambda_1 - \mathcal{O}\left( \frac{\varepsilon D}{d} \right)$, and $\hat{\lambda}_2 \leq \lambda_2$.

# D  Feedforward chain

In this Appendix, we derive the approximation for the learned dynamics matrix resulting from partial observations of a feedforward chain that we state in §3.2.

Suppose the teacher matrix has structure $B = QMQ^\top$ for $M_{ij} = \delta_{i+1,j}$, and $QQ^\top = Q^\top Q = I_D$. For convenience, we focus on the continuous-time limit. In this limit, the stationary covariance

$$\Sigma^\infty = \lim_{T \to \infty} \frac{1}{T} \int_0^T \mathbf{z}(t)\mathbf{z}(t)^\top dt$$

satisfies the relation

$$\Sigma^\infty = 2\sigma_\xi^2 \int_0^\infty e^{-(I_D-B)t} e^{-(I_D-B^\top)t} dt = 2\sigma_\xi^2 \int_0^\infty e^{-2t} e^{Bt} e^{B^\top t} dt$$

By the nilpotency of $B$, we have that

$$e^{Bt} = \sum_{n=0}^{D-1} \frac{(Bt)^n}{n!} = Q\left(\sum_{n=0}^{D-1} \frac{(Mt)^n}{n!}\right) Q^\top$$

$$\left(\sum_{n=0}^{D-1} \frac{(Mt)^n}{n!}\right)_{ij} = \begin{cases} \frac{t^{i-j}}{(i-j)!} & i \le j \\ 0 & i > j \end{cases}$$

We then have that

$$\left[e^{Mt} e^{M^\top t}\right]_{ij} = \sum_{k=\max(i,j)}^{D} = \frac{t^{2k-i-j}}{(k-i)!(k-j)!}$$

Defining $\Sigma^M$ as

$$[\Sigma^M]_{ij} = \int_0^\infty e^{-2t} \left[e^{Mt} e^{M^\top t}\right]_{ij} dt = \sum_{k=\max(i,j)}^{D} \frac{1}{2^{2k-i-j+1}} \binom{2k-i-j}{k-i},$$

we can express the stationary covariance as

$$\Sigma^\infty = 2\sigma_\xi^2 Q \Sigma^M Q^\top$$

The learned dynamics matrix is then given by

$$\hat{A} = PQM\Sigma^M Q^\top P^\top \left(PQ\Sigma^M Q^\top P^\top + \frac{\rho}{2\sigma_\xi^2} I_d\right)^{-1}$$

For simplicity, we consider the $Q = I_D$ case, with $\rho \to 0$. $\hat{A}$ will satisfy:

$$\hat{A}\left(P\Sigma^M P^\top\right) = PM\Sigma^M P^\top \tag{D.1}$$

Observe that $[P\Sigma^M P^\top]_{ij} = [\Sigma^M]_{ij}$ for $1 \le i, j \le d$, and that

$$[M\Sigma^M]_{ij} = \begin{cases} [\Sigma^M]_{i+1,j} & i \le D-1 \\ 0 & i = D \end{cases}$$

And thus, for $d < D$, $[PM\Sigma^M P^\top]_{ij} = [\Sigma^M]_{i+1,j}$ for $1 \le i, j \le d$. We can then make the ansatz that $\hat{A}_{ij} = \delta_{i+1,j} + \delta_{id}\hat{a}_j$ for some constants $\hat{a}_j$. This yields the following:

$$[\hat{A}\left(P\Sigma^M P^\top\right)]_{ij} = \begin{cases} [\Sigma^M]_{i+1,j} & i \le d-1 \\ \sum_{k=1}^{d} \hat{a}_k [\Sigma^M]_{kj} & i = d \end{cases}$$

The first $d - 1$ rows of D.1 are equal under this ansatz. The elements $\hat{a} \in \mathbb{R}^{1 \times d}$ are then chosen such that the $d^{th}$ row of D.1 matches, yielding that they must satisfy the following linear relation:

$$\sum_{k=1}^{d} \hat{a}_k [\Sigma^M]_{kj} = [\Sigma^M]_{d+1,j}, \quad 1 \le j \le d$$

$$\hat{a}[\Sigma^M]_{1:d,1:d} = [\Sigma^M]_{d+1,1:d}$$

Also note that $\hat{A}$ has the form of a companion matrix, and thus has eigenvalues given by the roots of the polynomial $f(\lambda) = \lambda^d - \sum_{k=1}^{d} \lambda^{k-1} \hat{a}_k$. Since $\hat{a} \ne 0$, we can say that $\hat{A}$ will have nonzero eigenvalues.

### D.1 Structure of the subsampled stationary covariance

When $d \ll D$ and $D$ is very large, $[P\Sigma^M P^\top]$ is well approximated as having a Toeplitz structure with constant differences between diagonals. Specifically, we claim that for $1 \le i, j \le d \ll D$,

$$[\Sigma^M]_{ij} = \sqrt{\frac{D}{\pi}} - \frac{|i-j|}{2} + \mathcal{O}\left(\frac{1}{\sqrt{D}}\right).$$

To show this, we must obtain asymptotics for

$$[\Sigma^M]_{ij} = \sum_{k=\max(i,j)}^{D} \frac{1}{2^{2k-i-j+1}} \binom{2k-i-j}{k-i}$$

when $1 \le i, j \le d$ as $D \to \infty$ for fixed $d$. It is easy to confirm that this sum is symmetric in $i$ and $j$, as

$$\binom{2k-i-j}{k-i} = \binom{(k-i)+(k-j)}{k-i} = \binom{2k-i-j}{k-j}.$$

Consider the lower triangular elements, letting $j = i - q$ for $q \in \{0, 1, 2, \ldots, i-1\}$. After shifting $k \leftarrow k - i$, we have

$$[\Sigma^M]_{i,i-q} = \sum_{k=0}^{D-i} \frac{1}{2^{2k+q+1}} \binom{2k+q}{k}.$$

It is then easy to see that the diagonal elements ($q = 0$) are weighted sums of central binomial coefficients:

$$[\Sigma^M]_{i,i} = \frac{1}{2} + \sum_{k=1}^{D-i} \frac{1}{2^{2k+1}} \binom{2k}{k}.$$

Then, using the bounds [64]

$$\frac{1}{2} \frac{4^k}{\sqrt{\pi k}} < \binom{2k}{k} < \frac{4^k}{\sqrt{\pi k}},$$

we have that

$$\frac{1}{2} + \frac{1}{4\sqrt{\pi}} \sum_{k=1}^{D-i} \frac{1}{\sqrt{k}} < [\Sigma^M]_{i,i} < \frac{1}{2} + \frac{1}{2\sqrt{\pi}} \sum_{k=1}^{D-i} \frac{1}{\sqrt{k}}.$$

Using asymptotics for generalized harmonic numbers [64], we have

$$\sum_{k=1}^{D-i} \frac{1}{\sqrt{k}} = 2\sqrt{D-i} + \mathcal{O}\left(\frac{1}{\sqrt{D-i}}\right).$$

For any fixed $i$, this immediately yields

$$[\Sigma^M]_{i,i} = \sqrt{\frac{D}{\pi}} + \mathcal{O}\left(\frac{1}{\sqrt{D}}\right).$$

Now, consider the off-diagonal elements, for $q \in \{1, 2, \ldots, i-1\}$. We remind ourselves that the sum of interest is

$$\sum_{k=0}^{D-i} \frac{1}{2^{2k+q+1}} \binom{2k+q}{k}$$

Using the recurrence

$$\binom{2k+q}{k} = \frac{2k+q}{k+q} \binom{2k+q-1}{k},$$

we have

$$\binom{2k+q}{k} \leq 2 \binom{2k+q-1}{k}$$

so

$$\sum_{k=0}^{D-i} \frac{1}{2^{2k+q+1}} \binom{2k+q}{k} \leq \sum_{k=0}^{D-i} \frac{1}{2^{2k+(q-1)+1}} \binom{2k+q-1}{k},$$

which shows that the matrix elements are non-increasing as one moves away from the diagonal:

$$[\Sigma^M]_{i,i-q} \leq [\Sigma^M]_{i,i-(q-1)}.$$

Moreover, we have from the same recurrence the weak lower bound

$$\binom{2k+q}{k} \geq \binom{2k+q-1}{k}$$

whence

$$[\Sigma^M]_{i,i-q} \geq \frac{1}{2}[\Sigma^M]_{i,i-(q-1)}.$$

These bounds show that all elements of the truncated covariance matrix must be of the same order. To show that the subleading term is of the desired form, we consider the difference between successive diagonals, which using the above identities may be expressed as

$$[\Sigma^M]_{i,i-(q-1)} - [\Sigma^M]_{i,i-q} = \sum_{k=0}^{D-i} \frac{1}{2^{2k+q+1}} \frac{q}{2k+q} \binom{2k+q}{k}.$$

Using the abovementioned bounds on central binomial coefficients, we have the bound

$$\frac{1}{2^{2k+q+1}} \frac{q}{2k+q} \binom{2k+q}{k} \leq \frac{1}{2^{2k+q+1}} \frac{q}{2k+q} \binom{2k+q}{k+q/2}$$

$$\leq \frac{1}{\sqrt{2\pi}} \frac{q}{(2k+q)^{3/2}}$$

which shows that the series is convergent as $D \to \infty$, with an $\mathcal{O}(1/\sqrt{D})$ remainder. In particular, letting $n = D - i + 1$, as this bound is monotone decreasing in $k$, we have

$$\sum_{k=n}^{\infty} \frac{1}{2^{2k+q+1}} \frac{q}{2k+q} \binom{2k+q}{k} \leq \sum_{k=n}^{\infty} \frac{1}{\sqrt{2\pi}} \frac{q}{(2k+q)^{3/2}}$$

$$\leq \frac{1}{\sqrt{2\pi}} \frac{q}{(2n+q)^{3/2}} + \int_{n}^{\infty} dk \, \frac{1}{\sqrt{2\pi}} \frac{q}{(2k+q)^{3/2}}$$

$$= \frac{1}{\sqrt{2\pi}} \frac{q}{(2n+q)^{3/2}} + \frac{1}{\sqrt{2\pi}} \frac{q}{(2n+q)^{1/2}}$$

$$= \mathcal{O}\left(\frac{1}{\sqrt{D}}\right).$$

What remains is to compute the infinite sum, which evaluates to

$$\sum_{k=0}^{\infty} \frac{1}{2^{2k+q+1}} \frac{q}{2k+q} \binom{2k+q}{k} = \frac{1}{2}$$

for $q \geq 1$. Therefore, we have

$$[\Sigma^M]_{i,i-(q-1)} - [\Sigma^M]_{i,i-q} = \frac{1}{2} + \mathcal{O}\left(\frac{1}{\sqrt{D}}\right),$$

hence in combination with our previous result for the diagonal terms we conclude that

$$[\Sigma^M]_{i,i-q} = \sqrt{\frac{D}{\pi}} - \frac{q}{2} + \mathcal{O}\left(\frac{1}{\sqrt{D}}\right),$$

or, restoring the indices, we obtain the claimed result that

$$[\Sigma^M]_{ij} = \sqrt{\frac{D}{\pi}} - \frac{|i-j|}{2} + \mathcal{O}\left(\frac{1}{\sqrt{D}}\right).$$

This shows that the subsampled stationary covariance matrix is approximately Toeplitz.

### D.2   Structure of the student dynamics matrix under heavy subsampling

Now, we consider the structure of the student's dynamics matrix in the $d \ll D$ regime. The inverse of the form of Toeplitz matrix by which the stationary covariance is approximated is known to take the form [65]:

$$[P\Sigma^M P^\top]^{-1} \approx \begin{bmatrix} 1 - \frac{1}{\mathcal{O}(c)+\mathcal{O}(d)} & -1 & 0 & \cdots & 0 & \frac{1}{\mathcal{O}(c)+\mathcal{O}(d)} \\ -1 & 2 & -1 & 0 & \cdots & 0 \\ 0 & -1 & 2 & -1 & 0 & \cdots \\ \vdots & \ddots & \ddots & \ddots & \ddots & \vdots \\ 0 & 0 & 0 & -1 & 2 & -1 \\ \frac{1}{\mathcal{O}(c)+\mathcal{O}(d)} & 0 & 0 & 0 & -1 & 1 - \frac{1}{\mathcal{O}(c)+\mathcal{O}(d)} \end{bmatrix}$$

where $c = \sqrt{D/\pi}$. We also have that $PM\Sigma^M P^\top \approx P\Sigma^M P^\top + \frac{1}{2}R$ where $R_{ij} = \mathbb{1}(i < j) - \mathbb{1}(i \geq j)$. Thus,

$$\hat{A} = PM\Sigma^M P^\top \left(P\Sigma^M P^\top\right)^{-1} \approx I_d + \frac{1}{2}R[P\Sigma^M P^\top]^{-1}.$$

Taking the large $c$ approximation, we find that the learned student dynamics approaches the form

$$\hat{A}_{ij} = \delta_{i+1,j} + \delta_{id}\delta_{ij}.$$

In other words, $\hat{A}$ approaches a feedforward chain of size $d$, except with the activity of the start of the chain never decaying. The largest learned eigenvalue in this limit is 1, while the others vanish identically.

We note that in practice, the sensitivity of the eigenvalues of feedforward chain connectivity matrices to small perturbations would cause multiple of the learned eigenvalues to be significantly larger than 0. In particular, the $\varepsilon$-pseudospectrum of a feedforward chain of length $d$ has a radius on the order $\varepsilon^{1/d}$ [66].

## E   Low rank

In this Appendix, we derive the results on MAP inference for low-rank null teachers stated in §3.3.

Consider a low-rank teacher of the form $B = MN^\top$, $M \in \mathbb{R}^{D \times r}$, $N \in \mathbb{R}^{D \times r}$. If $N^\top M = \mathbf{0}_{r \times r}$ and $N^\top N = M^\top M = \gamma^2 I_r$, then $B$ has all 0 eigenvalues, but is nonnormal. Here $\gamma^2$ is a scale parameter, which in some sense controls the degree of non-normality (scales the norm of the

commutator $[B, B^\top] = BB^\top - B^\top B$). We compute the stationary covariance of the teacher process, suppressing factors of $\sigma_\xi^2$ by setting $\sigma_\xi^2 = 1$:

$$\Sigma^\infty = 2 \int_0^\infty e^{-(I_D - B)t} e^{-(I_D - B^\top)t} dt = \int_0^\infty e^{-2t} e^{MN^\top t} e^{NM^\top t} dt$$

$$= 2 \int_0^\infty e^{-2t} \exp\left(t \sum_{i=1}^r m_i n_i^\top\right) \exp\left(t \sum_{k=1}^r n_k m_k^\top\right) dt$$

Observe that $m_i n_i^\top$ commutes with $m_j n_j^\top$ due to the $N^\top M = \mathbf{0}$ constraint. Thus, we can write

$$\Sigma^\infty = 2 \int_0^\infty e^{-2t} \prod_{i=1}^r \exp\left(m_i n_i^\top t\right) \prod_{k=1}^r \exp\left(n_k m_k^\top t\right) dt$$

$$= 2 \int_0^\infty e^{-2t} \prod_{i=1}^r (I_D + m_i n_i^\top t) \prod_{k=1}^r (I_D + n_k m_k^\top t) dt$$

$$= 2 \int_0^\infty e^{-2t} (I_D + Bt)(I_D + B^\top t) dt$$

$$= 2 \int_0^\infty e^{-2t} (I_D + 2B_s t + BB^\top t^2) dt$$

where $B_s = \frac{B + B^\top}{2}$. Performing this integral yields the solution

$$\Sigma^\infty = I_D + B_s + \frac{1}{2} BB^\top.$$

## E.1 Spectrum of the stationary covariance

Our first goal is to determine the eigenvalues and eigenvectors of $\Sigma^\infty$. To do so, suppose that $\mathbf{u} \in \mathbb{R}^D$ is a unit-norm eigenvector of $\Sigma^\infty$ with eigenvalue $\lambda$. Then, it must satisfy

$$\Sigma^\infty \mathbf{u} = \mathbf{u} + \frac{1}{2} MN^\top \mathbf{u} + \frac{1}{2} NM^\top \mathbf{u} + \frac{1}{2} \gamma^2 MM^\top \mathbf{u} = \lambda \mathbf{u}.$$

As $M$ and $N$ span orthogonal $r$-dimensional subspaces of $\mathbb{R}^D$, one possibility is that $\mathbf{u}$ lies in the $(D - 2r)$-dimensional complement of those subspaces, in which case it must have eigenvalue 1. Thus, $\Sigma^\infty$ has eigenvalue 1 with multiplicity $D - 2r$. Now consider the case in which $\mathbf{u}$ lies in the union of the subspaces spanned by $M$ and $N$. Make a decomposition

$$\mathbf{u} = M\mathbf{a} + N\mathbf{b},$$

where $\mathbf{a}, \mathbf{b} \in \mathbb{R}^r$. The unit-norm condition is

$$1 = \|\mathbf{u}\|^2 = \gamma^2 (\|\mathbf{a}\|^2 + \|\mathbf{b}\|^2),$$

while the eigenvector condition becomes

$$\Sigma^\infty \mathbf{u} = M\mathbf{a} + N\mathbf{b} + \mathbf{u}_\perp + \frac{1}{2} M\gamma^2 \mathbf{b} + \frac{1}{2} N\gamma^2 \mathbf{a} + \frac{1}{2} \gamma^4 M\mathbf{a}$$

$$= \lambda [M\mathbf{a} + N\mathbf{b} + \mathbf{u}_\perp].$$

Acting with $M^\top$, we have

$$\mathbf{a} + \frac{1}{2} \gamma^2 \mathbf{b} + \frac{1}{2} \gamma^4 \mathbf{a} = \lambda \mathbf{a}$$

while acting with $N^\top$, we have

$$\mathbf{b} + \frac{1}{2} \gamma^2 \mathbf{a} = \lambda \mathbf{b}.$$

Together these conditions imply that $\mathbf{b} = t\mathbf{a}$, which gives a coupled set of equations for $t$ and $\lambda$:

$$1 + \frac{1}{2}\gamma^2 t + \frac{1}{2}\gamma^4 = \lambda$$

$$t + \frac{1}{2}\gamma^2 = \lambda t.$$

This linear system has solutions

$$\lambda_\pm = \frac{4 + \gamma^4 \pm \gamma^2\sqrt{4 + \gamma^4}}{4}$$

$$t_\pm = \frac{-\gamma^2 \pm \sqrt{4 + \gamma^4}}{2},$$

which each must correspond to orthogonal $r$-dimensional eigenspaces. Therefore, we at last conclude that the eigenvalues of $\Sigma^\infty$ are 1 with multiplicity $D - 2r$ and $\lambda_\pm$, each with multiplicity $r$. When $\gamma \gg 1$, this gives an $r$-dimensional 'signal' eigenspace with eigenvalue

$$\lambda_+ = \frac{4 + \gamma^4 + \gamma^2\sqrt{4 + \gamma^4}}{4} = \frac{\gamma^4}{2} + \frac{3}{2} + \mathcal{O}\left(\frac{1}{\gamma^4}\right),$$

a $(D - 2r)$-dimensional 'null' eigenspace with eigenvalue 1, and an $r$-dimensional 'suppressed' eigenspace with eigenvalue

$$\lambda_- = \frac{4 + \gamma^4 - \gamma^2\sqrt{4 + \gamma^4}}{4} = \frac{1}{2} + \mathcal{O}\left(\frac{1}{\gamma^4}\right).$$

As a result, increasing $\gamma$ will push the effective dimensionality of activity in the stationary state closer to $r$.

## E.2  Spectrum of the learned dynamics matrix for large $\gamma$

We now turn to our main goal, which is to approximately determine the eigenvalues of the learned dynamics matrix after subsampling. Using our result for the stationary covariance, we find that the learned dynamics matrix in the infinite time limit is given by

$$\hat{A} = PB(I_D + B_s + \frac{1}{2}BB^\top)P^\top(P(I_D + B_s + \frac{1}{2}BB^\top)P^\top)^{-1}$$

$$= (\tilde{M}\tilde{N}^\top + \frac{\gamma^2}{2}\tilde{M}\tilde{M}^\top)\left(I_d + \frac{\tilde{M}\tilde{N}^\top + \tilde{N}\tilde{M}^\top}{2} + \frac{\gamma^2}{2}\tilde{M}\tilde{M}^\top\right)^{-1}$$

where $\tilde{M} = PM$ denotes $M$ truncated to the first $d$ rows. Since $MN^\top$ is of rank $r$, $\hat{A}$ will have at most $r$ non-zero eigenvalues.

The relevant regime is when $\gamma \gg 1$, such that the activity is approximately low-dimensional. Because of the normalization condition $N^\top N = M^\top M = \gamma^2 I_r$, in any fixed dimension we must have $N_{ij} = \mathcal{O}(\gamma)$, $M_{ij} = \mathcal{O}(\gamma)$. We can then consider making $\gamma$ parametrically large, in which case we have

$$\hat{A} = \Pi_{\tilde{M}\tilde{M}^\top} + \mathcal{O}\left(\frac{1}{\gamma^2}\right)$$

where $\Pi_{\tilde{M}\tilde{M}^\top}$ is the orthogonal projector onto the $r$-dimensional span of $\tilde{M}\tilde{M}^\top$. Here, we have used the fact that $\gamma^2\tilde{M}\tilde{M}^\top \sim \mathcal{O}(\gamma^4)$ and $\tilde{M}\tilde{N}^\top \sim \mathcal{O}(\gamma^2)$, so the former terms will dominate at large $\gamma$. Therefore, it follows that as $\gamma$ becomes large the $r$ non-zero eigenvalues of $\hat{A}$ tend to one. This argument relies on fixing all dimensions.

A case of interest is when $\gamma^2 \sim \mathcal{O}(D/\sqrt{r})$ for $D \gg r$; with this scaling, the elements of $B$ are $\mathcal{O}(1)$ with respect to $D$ and $r$.

### E.3  Low rank teachers with nontrivial eigenspectra

While the null overlap case is analytically tractable, we would like to also consider non-normal low rank teachers with nontrivial eigenspectra. To do this, we can specify an overlap matrix $Q$ with the desired spectra, such that $N^\top M = Q$. For a fixed $M$ (e.g. selected with random entries), we can satisfy the desired overlap via selecting $N^\top$ as $N^\top = QM^\dagger$ where $M^\dagger = M^+ + V(I_D - MM^+)$ is a generalized pseudoinverse. Here, $V \in \mathbb{R}^{r \times D}$ is arbitrary, and $M^+$ is the Moore-Penrose pseudoinverse. Observe that in the case where $M$ has orthogonal columns and $V = \mathbf{0}_{r \times D}$, $M^\dagger = M^+ = M^\top$, yielding that $B = MN^\top$ is normal so long as $Q$ is normal.

Thus, there are then two ways to make $B = MN^\top = MQM^\dagger$ non-normal; one is to choose choose $V = \mathbf{0}_{r \times D}$ and make $Q$ non-normal. Another way is to select nonzero $V$, e.g. with entries drawn from $\mathcal{N}(0, \sigma_V^2)$. Then, one can construct a highly non-normal $B$ even with a normal overlap matrix. We use the latter method to generate non-normal low rank teacher connectivity that supports oscillatory dynamics. Specifically, for Fig. 5, we construct teacher connectivity by generating $M$ with random entries $\sim \mathcal{N}(0, 1)$, $Q = \begin{pmatrix} 0.5 & -0.5 \\ 0.5 & 0.5 \end{pmatrix}$ and $\sigma_V = \frac{12}{\sqrt{D}}$ for $D = 500$.

## F  Regularization

In the analysis thus far, we have focused on learning the dynamics matrix in the ridgeless limit. However, using a high enough regularization may help prevent the discovery of spurious slow modes of dynamics from non-normal transients. Indeed, this is necessarily true by observing that in the opposing limit $\rho \uparrow \infty$, the learned dynamics matrix collapses to the zeros matrix. One can attempt to estimate an optimal regularization parameter that yields a predictive fit via cross-validation, but it is unclear whether this would necessarily resolve the eigenvalue overstimation issues described here. Obtaining an accurate estimate of the optimal ridge from cross-validation on windows of correlated timeseries data is statistically challenging [67], and ill-defined if the timeseries is non-stationary. Furthermore, such problems arising from an inherent expressivity issue of the student network—such as when fitting a low-dimensional LDS model to a functionally feedforward chain—clearly cannot simply be resolved through regularization. We leave a more detailed investigation of regularization-based strategies for ablating spuriously discovered attractor-like dynamics to future work.

## G  Numerical methods and supplemental figures

All of our numerical simulations are implemented in `Python` 3.9.18 using `NumPy` 1.26.2 [68], `SciPy` [69], and `PyTorch` [70]. They were not computationally-intensive, and required less than 12 hours in total to run on a consumer Dell XPS laptop equipped with an Intel Core™ i7-13700H processor. Code to reproduce all experiments is available at `https://github.com/wqian0/DataConstrainedRNNs/`.

For simplicity, we use $\tau = 1$ in all numerical simulations. Unless stated otherwise, we use a teacher network size of $D = 500$ in all numerical experiments.

We integrate the student and teacher RNN dynamics via Euler integration with a timestep $\Delta t = 0.01$. Under the discretization scheme of B.4, in all experiments, we select the noise parameters of the student and teacher dynamics as $\sigma_\eta = \sigma_\xi = \frac{0.02}{\sqrt{2}}$.

In the examples of Fig. 1, we generate ground truth network activity by iterating the dynamics for a duration $T = 5000 \times \Delta t$.

In the purely noise-driven experiments with finite observation time windows, we fit student networks to ground truth teacher activity generated over a duration $T = 30000 \times \Delta t$.

For MAP inference, we use a regularization parameter $\rho = 0.001$ in all experiments. In experiments involving the long time limit $T \to \infty$, we use `SciPy` [69]'s built-in Lyapunov solver to compute the stationary covariance of the teacher activity.

For all LDS models, we run the fitting procedure for 200 iterations using the implementation provided by the authors of [18] under an MIT License on GitHub.[2] For the experiments in Fig. 1, the input

---
[2]`https://github.com/lindermanlab/ssm`

signal was explicitly passed to the fitting procedure. For the purely noise-driven teacher setting of Fig. G.5, no input signal was provided.

Fig. 5: We note that CORNN is applicable only for fitting RNNs that obey slightly different dynamics (leaky-rate instead of leaky-current), of the form

$$\tau \dot{\mathbf{x}} = -\mathbf{x} + \phi(A\mathbf{x}) + \boldsymbol{\eta}(t)$$

We thus modify student and teacher accordingly in experiments involving CORNN [14]. We run the CORNN fitting procedure for 1000 iterations using the implementation provided by the authors as free software on GitHub.[3]

For backpropagation-through-time, we train student network dynamics matrices on observed teacher activity via truncated BPTT, splitting the observed 30000 timepoints into segments of 500 contiguous timepoints. Student networks were then trained for 100 epochs at a learning rate 0.01, batch size of 16, and a teacher forcing ratio of 0.5.

For experiments with FORCE, we train student networks for 1000 iterations with parameters $g = 1.5$, $P_0 = 0.01$. We adapt the implementation provided by the authors of [17] under a GNU GPL license on Github.[4] To increase agreement of the fitted activity, we restricted the observation window of the fitted network to the first 5000 timepoints; fitting to the full observation window yielded fits that failed to capture prominent initial transients of the observed activity.

For experiments involving low-rank teacher networks with null overlap, we use $\gamma^2 = 0.2 \frac{D}{\sqrt{r}}$ for the linear networks (Fig. 4) and $\gamma^2 = 0.4 \frac{D}{\sqrt{r}}$ for the nonlinear networks (Fig. G.6). This scaling ensures elements of $B$ are order-1 (see E.2). For low-rank teacher networks with nontrivial eigenspectra (Fig. 5), see E.3.

---

[3]https://github.com/schnitzer-lab/CORNN-public
[4]https://github.com/rajanlab/CURBD

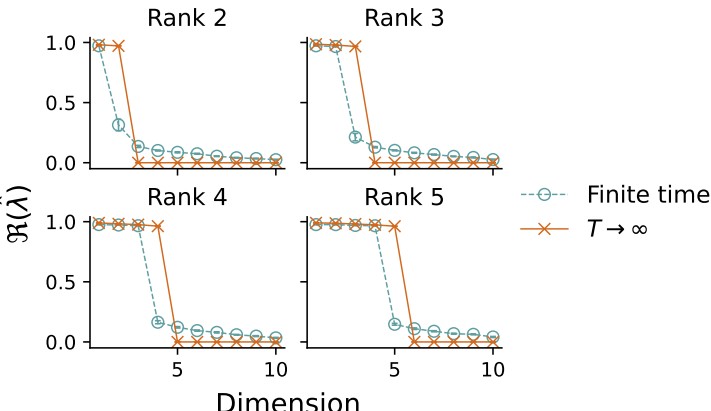

Figure G.1: Finite time effects for learning from low-rank teachers with null overlap connectivity. The top ten largest learned eigenvalues are shown for students learning from teachers of rank $2 \leq r \leq 5$ and null overlap connectivity (all 0 eigenspectrum). All plots correspond to connectivity parameter $\gamma^2 = D/\sqrt{r}$ with a teacher size $D = 500$ and 5% partial observation. Each point indicates an average over 20 randomly selected teacher networks. Error bars are $\pm 1$ standard error of the mean, but are mostly too small to see.

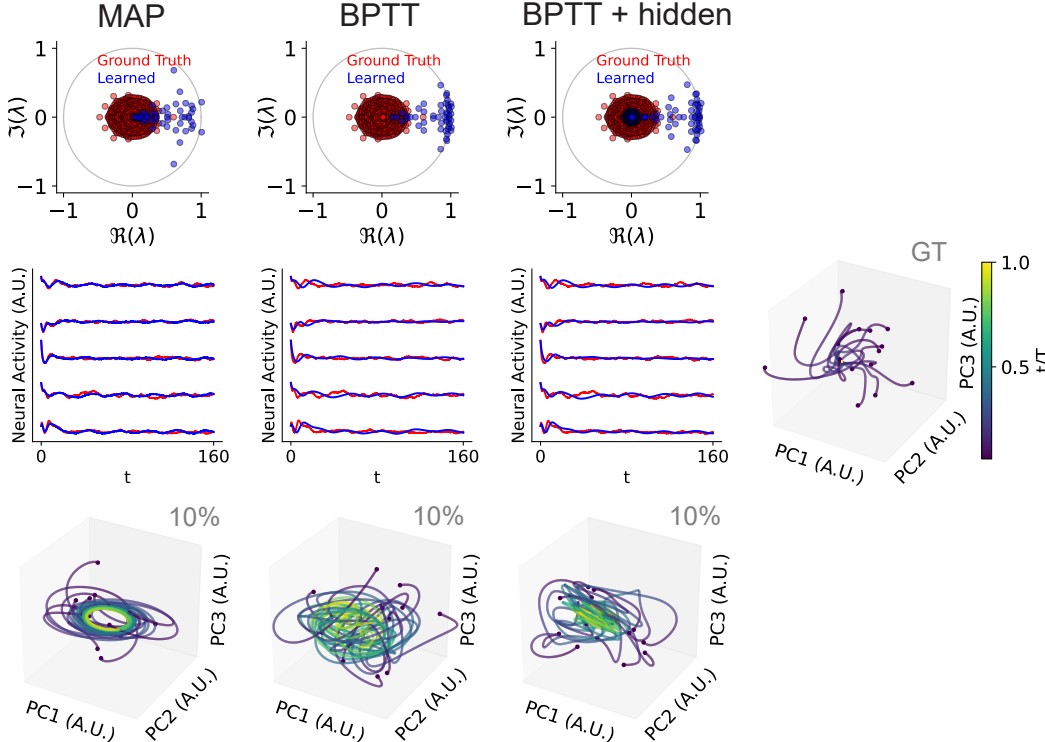

Figure G.2: Eigenvalue estimation under partial observation extends more generally to unstructured non-normal teacher connectivity, and occurs even when hidden neurons are added during training. Each column corresponds to an inference method (MAP, BPTT, BPTT with 200 additional hidden neurons) at 10% partial observability. Top row: ground truth teacher (red) and learned student (blue) dynamics matrix eigenvalues. Middle row: Activity traces for the teacher (red) and student (blue) networks. Bottom row: Example student network dynamics, as shown by randomly sampled initial conditions. Right: Ground truth dynamics.

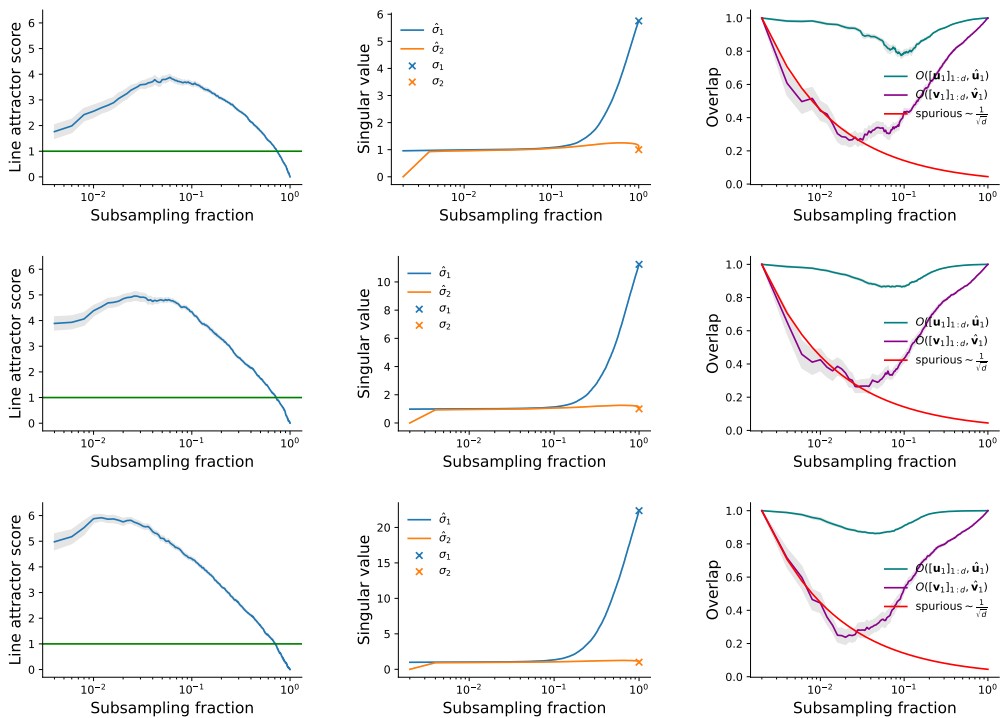

Figure G.3: Properties of learned student dynamics matrices for functionally feedforward teachers in the long time limit $T \to \infty$. Each row corresponds to a functional skip connection strength $\beta \in \{0.25, 0.5, 1\}$. Left: Line attractor score versus subsampling fraction ($d/D$). The green line indicates a line attractor score of 1. Middle: Top two singular values of the learned (student) and true (teacher) dynamics matrices as a function of subsampling fraction. Right: Normalized overlap (absolute cosine similarity) of the learned left and right singular vectors corresponding to the largest learned singular value ($\hat{u}_1$, $\hat{v}_1$, respectively) with the truncated top left and right singular vectors of the true network ($[u_1]_{1:d}$, $[v_1]_{1:d}$, respectively). The red curve shows how the expected overlap would approximately scale for arguments with randomly selected entries. All plots show averages over 20 randomly selected teacher networks. The shaded regions indicate $\pm 1$ standard error of the mean, and is in some cases too small to see.

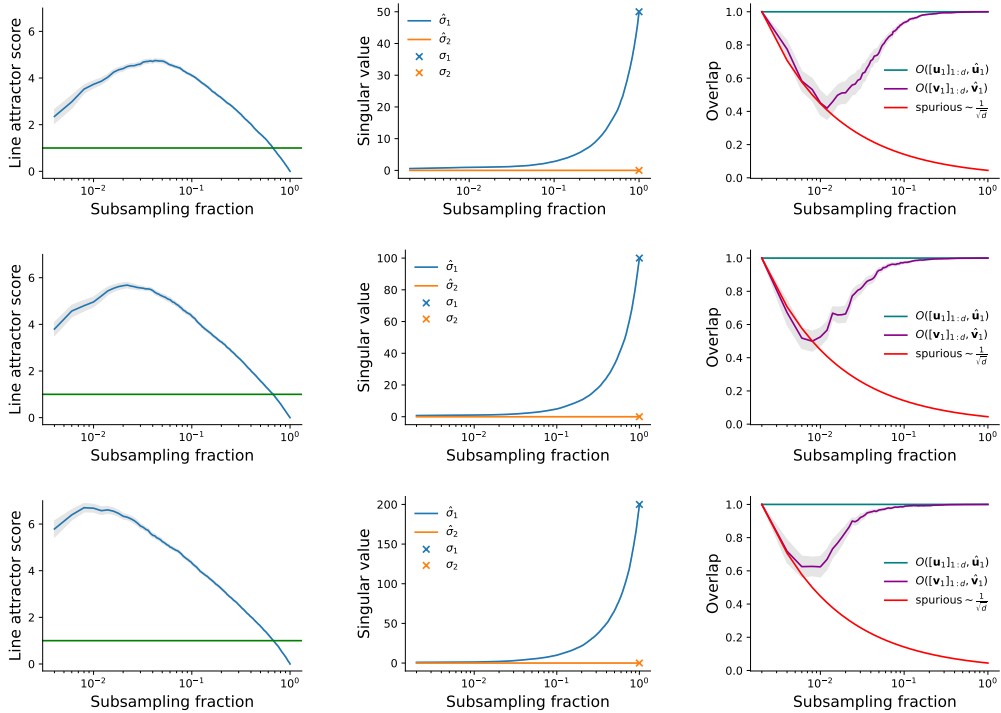

Figure G.4: Properties of learned student dynamics matrices for rank-1 teachers with null overlap connectivity in the long time limit $T \to \infty$. Each row corresponds to a different value of $\gamma^2$ ($\gamma^2 = \{0.1, 0.2, 0.4\}D$). Left: Line attractor score versus subsampling fraction ($d/D$). The green line indicates a line attractor score of 1. Middle: Top two singular values of the learned (student) and true (teacher) dynamics matrices as a function of subsampling fraction. Right: Normalized overlap (absolute cosine similarity) of the learned left and right singular vectors corresponding to the largest learned singular value ($\hat{u}_1$, $\hat{v}_1$, respectively) with the truncated top left and right singular vectors of the true network ($[u_1]_{1:d}$, $[v_1]_{1:d}$, respectively). The red curve shows how the expected overlap would approximately scale for arguments with randomly selected entries. All plots show averages over 20 randomly selected teacher networks. The shaded regions indicate $\pm 1$ standard error of the mean, and is in some cases too small to see.

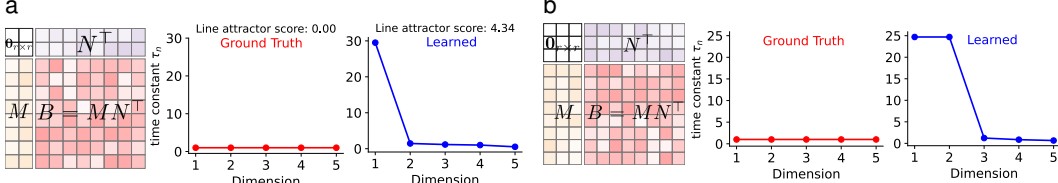

Figure G.5: LDS fits to the low-rank linear teacher networks considered in Fig. 4 (**a.** rank 2 and **b.** rank 3, respectively). Left: A schematic of teacher connectivity. Middle: The top five time constants of the ground truth teacher connectivity. Right: The time constants learned by an LDS model fit to the teacher activity. All fits were performed at 5% partial observability.

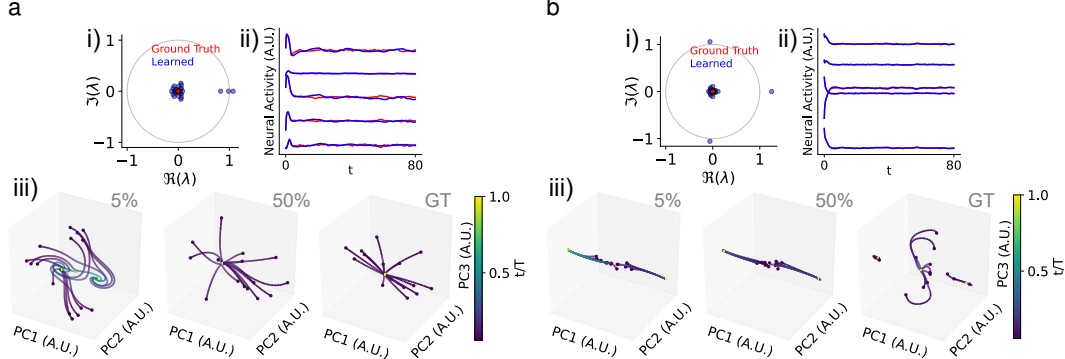

Figure G.6: Spurious or missed stable fixed points learned from rank-3 teachers with null overlap connectivity in the nonlinear setting. **a.** i). Ground truth teacher (red) and learned student (blue) dynamics matrix eigenvalues at 5% subsampling. ii). Activity traces for the teacher (red) and student (blue) networks at 5% subsampling. iii). Example student network dynamics for 5% and 50% subsampling compared to the ground truth (GT).**b.** As in **a**, but for another example teacher network with rank 3 null overlap connectivity.

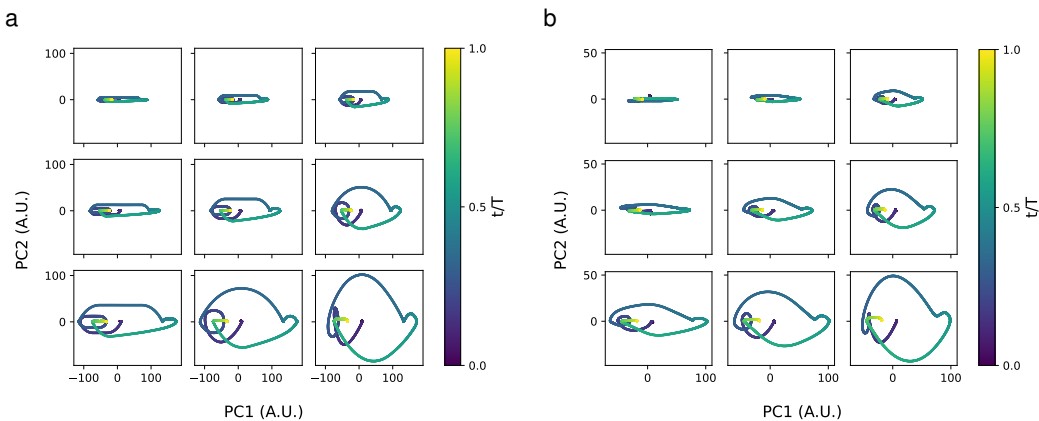

Figure G.7: Delay embeddings for integrator circuits performing the integration task of Fig. 1. **a.** Top two PCs of delay-embedded observed activity from the line attractor performing the integration task. Each row corresponds to a different number of delays $\{5, 10, 20\}$, and each column corresponds to a delay interval in $\{5, 10, 20\}\Delta t$. **b.** As in **a**, but for the functionally feedforward chain.

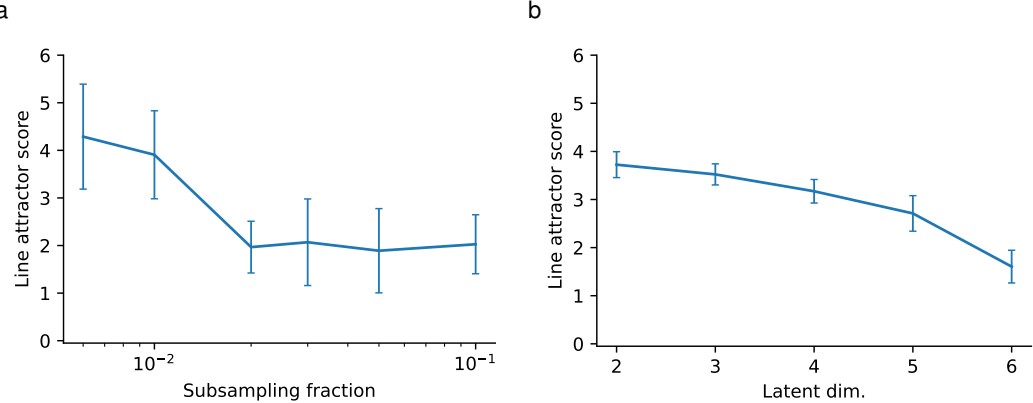

Figure G.8: Sweep of LDS models fit to a functionally feedforward chain performing the integration task of Fig. 1. **a.** Line attractor score versus subsampling fraction for LDS models fit with maximal latent dimension $d - 1$. **b.** Line attractor score versus LDS latent dimension at full observability $(d = D)$. In both plots, each point represents an average over ten fits.

