# OpenReview forum: "Partial observation can induce mechanistic mismatches in data-constrained models of neural dynamics"
_NeurIPS.cc/2024/Conference — NeurIPS 2024 poster_

### Official Review · Reviewer_Tqmw · 2024-07-10

**Soundness:** 3
**Presentation:** 3
**Contribution:** 3
**Rating:** 7
**Confidence:** 5

**Summary:**

The authors investigate whether dynamical systems models that are statistically fit to neural data recapitulate the same dynamical mechanisms of the circuit. Commonly, neuroscientists only measure activity from a small fraction of neurons within a circuit. Under this constraint, the authors show examples where the model learns fundamentally different dynamical mechanisms. If true, this would be of great significance to the computational neuroscience community and an important cautionary tale.

**Strengths:**

There are a lot of good things to say about this paper. The motivating question is very interesting and important. The mathematical framing and analysis is elegant. The introduction and discussion sections are skillfully written to target the computational neuroscience community, and I think the paper has the potential to be very impactful within this community after some revisions.

**Weaknesses:**

I very much liked the big picture idea of this paper. However, in various parts of the paper, the writing is unclear and left me confused on details. The interpretation of certain findings should be clarified. I am optimistic about being to raise my score during the discussion period if these weaknesses can be addressed.

* In section 2, the authors fit a linear dynamical system model with 5 latent dimensions (line 113) to 5% of neurons sampled from a feedforward chain circuit. The model fails to recapitulate the feedforward chain and instead learns a line attractor. Critically, there are two distinct but non-mutually exclusive explanations for this result: (a) 5 latent dimensions is not enough to capture the line attractor, or (b) the partial observation of 5% of the circuit is insufficient. The title and abstract of the paper exclusively favor explanation (b). However, on lines 127-132 the authors do not mention (b) and instead put forth (a), "[if] the data constrained model has fewer neurons, it cannot realize a feedforward chain with sufficiently long memory". Here, the recurrent dynamics in the LDS happen within a 5-dimensional latent space, so I assume that the authors mean "fewer neurons" to mean the relatively low dimensionality of the latent space.

* I would like to see the result in section 2 more systematically investigated as follows. Let `N` denote the number of neurons in the teacher circuit (feedforward chain). Let `d` denote the latent dimensionality of the LDS model, and let `n` denote the number of neurons that are measured / observed for fitting the LDS model. I am interested to see whether the LDS model still fails to learn a feedforward chain when `N` is large (e.g. maybe 1000) and `n = d` is big but still a small fraction of the overall system (e.g. `n = d = 100`). Additionally, it would be elucidating to see an example where `N = n` (i.e. fully observed circuit), but a low-dimensional dynamical model is assumed `d < n`. The purpose of these additional experiments is to clarify whether the problem is that `n` $\ll$ `N` (as suggested by the title / abstract), or whether the real problem is that `d` $\ll$ `N`, or whether both are a problem.

* Figure 3 seems to get at some of these concerns, but I think the paper will be much clearer if the above comments are addressed in Figure 1 as well.

* In Figure 3, it is still a little unclear to me whether subsampling, per se, is the problem. If I understand the construction of the teacher network properly, as `D` increases, the timescale of the feedforward chain also increases. It is harder for the student network to match this timescale with a chain, leading it to instead learn a line attractor. I would like to see an experiment where the timescale of the feedforward chain is kept constant while increasing `D` (e.g. by inserting multiple copies of each neuron in the feedforward chain). In this case, when you randomly sample `d` neurons for the student network, do the learned dynamics still fail to capture a feedforward chain for large `D`? An affirmative answer would seem to be more in agreement with the main claims of the title and abstract. A negative answer would still be interesting, but would change the interpretation of the paper to be more in line with the idea that a mismatch in dimensionality is the core of the problem instead of sub-sampling.

**Questions:**

See "Weaknesses".

**Limitations:**

Limitations are adequately discussed.

---

> ### Author Rebuttal · Authors · 2024-08-06
>
> We thank the reviewer for their careful critiques of our work, which have helped us further strengthen our paper. Below we respond to their comments on Weaknesses of our work:
> * The reviewer brings up an excellent point---that both partial observation and the selection of a latent dimension much smaller than the number of observed neurons could contribute to why the feedforward chain in the experiment of Section 2 is misidentified as a line attractor. We set up the experiment to parallel how data-constrained models are often fit in practice, with both partial observation and an explicit bias towards smaller latent dimensions, but appreciate the concern that this leaves room for multiple explanations as to why the mismatch we found occurs. However, since partial observation even in the absence of an explicit bias towards small latent dimensions already bounds model dimension to be smaller than that of the teacher network, the quoted explanation we provide was intended to implicitly also support explanation (b). Nonetheless, we will provide further clarification on this point and our interpretation of the findings here.
> * We agree that the experiment you propose would be fruitful in disentangling explanations (a) and (b), and as such have performed a sweep of LDS models of various latent dimensions and observability conditions, fit to the same functionally feedforward chain as in Fig. 1b.
>
>     In particular, using the suggested notation, we fit models with a latent dimension $d = n-1$ (the $-1$ accounts for the fact that the LDS fitting procedure we use requires latent dimension to be strictly less than the observation dimension) to a functionally feedforward teacher circuit performing the same integration task (as in Fig. 1b). We performed fits at various values of $n$, up to the suggested fraction of $n = 0.1N$. In addition, as suggested, we fit LDS models of small latent dimensions $d \in \{2, 3, \dots, 6\}$ at full observability.
>
>     The results demonstrate that the mechanistic mismatch we observe persists even if partial observability is the only driver of restricted model dimension. Further, the second experiment demonstrates that  a model of small latent dimension will still learn a line attractor-like mechanism even at full observability. This finding is consistent with the quoted explanation of lines 127-132.
>
> * We agree that an experiment with a fixed length feedforward chain while increasing $D$ could further clarify how the results of Fig. 3 should be interpreted. We interpret the suggested connectivity for such an experiment proposed by the reviewer as resembling synfire chain connectivity. Specifically, we consider teacher networks with connectivity $B = \frac{1}{k}\delta_{i+1,j} \otimes \mathbf{1}_{k\times k}$, where $k$ is the number of copies of each neuron in the chain. Here, the $\frac{1}{k}$ factor is included to ensure that all the nonzero singular values of $B$ remain equal to $1$ as the number of copies $k$ is scaled up. We compute the learned student matrix for a random selection of $d=50$ observed neurons in the limit of long observation time $T$, for fixed chain length $D/k = 50$ and values of $k \in \{2^0, 2^1, \dots 2^6\}$. For each value of $k$, we report the top two eigenvalues, time constants, and singular values of the learned student networks, averaged over $10$ random observed neuron selections, sampled without replacement.
>
>     The results are consistent with the interpretation that neither a feedforward chain, nor a line attractor (or any persistent timescale mechanism) are learned when $D$ is increased in this manner. Specifically, as $k$ is increased, the learned timescales approach the intrinsic neuronal timescale $\tau$, ruling out persistent timescale mechanisms, and the learned singular values fall well below $1$, ruling out a feedforward chain-like structure. Thus, as in Fig. 3, the student still does not learn a feedforward chain, but fails to do so in a different manner.
>
> Plots for the latest experiments are included in the global response.

---

> > ### Comment · Reviewer_Tqmw · 2024-08-09
> > **Thanks for the additional experiments**
> >
> > Thanks for the additional experiments. My interpretation of them is that both neuron subsampling and latent dimension restriction are problems for creating a dynamical mismatch. It would be great to see follow up work on this that carefully disentangles these two effects further. I'm raising my score to a 6, as I think the paper does a good first pass characterization of this important phenomenon.

---

> > > ### Author Response · Authors · 2024-08-09
> > >
> > > Thank you for your prompt response; we are glad that the additional experiments addressed some of your concerns. We agree that both subsampling and latent dimensionality restriction contribute towards the bias towards line-attractor-like mechanisms, and certainly intend to follow up on our findings in future work. As you say, this is just the first step!

---

### Official Review · Reviewer_VFVu · 2024-07-11

**Soundness:** 3
**Presentation:** 4
**Contribution:** 3
**Rating:** 6
**Confidence:** 4

**Summary:**

The authors address provide a cautionary tale for how data-constrained RNN models can mis-identify the dynamical structure underlying neural population computations when only a subset of the neural population is observed ("partial observation").  Empirical case studies are provided, whereby data-constrained student networks are trained from partial observation of synthetic teacher networks. The analyzed dynamics of the trained student networks differed from those of the teacher network in several teacher network setups. When teacher networks performed integration of a stimulus input via a feedforward chain dynamic, student networks were found to be biased to recover a line attractor dynamic under certain conditions. Mismatched student-teacher dynamics were also shown when teachers were linear noise-driven networks with non-normal dynamics or low-rank connectivity. The final case study provided details similar propensities for mismatched student-teacher dynamics when all networks are non-linear.

**Strengths:**

- Originality: The overall message and cautionary tale is indeed original (although see related work in Weaknesses).

- Quality: The case study examples are well chosen and illustrative of the potential for mismatched student-teacher dynamics. The experiments appear well controlled and visualizations are intuitive and elegant.

- Clarity: Overall, the writing is clear and appropriately didactic.

- Significance: Recent years have seen a proliferation of data-constrained model development and interest in interpreting the corresponding trained models. Thus, this cautionary tale message is quite significant and timely. While perhaps beyond the scope of this work, solutions to the problems exposed, even if only hypothesized, could improve the significance of this work.

**Weaknesses:**

- Minor: There is a line of related work from Chethan Pandarinath's group on interpretable data-constrained models that should be referenced if not explicitly discussed. For example: Sedler, Andrew R., Christopher Versteeg, and Chethan Pandarinath. 2023. “Expressive Architectures Enhance Interpretability of Dynamics-Based Neural Population Models.” Neurons, Behavior, Data Analysis, and Theory, March, 1–22. https://doi.org/10.51628/001c.73987.

**Questions:**

- If I understood correctly, the student models applied throughout were designed to provide a one-to-one mapping from hidden units in the student RNN to observed teacher unit activity. Other data-constrained architectures do not rely on such design constraints--in particular, LFADS [11] and Neural ODEs [33] attempt to reproduce single-trial neural recordings (or teacher activity) via a readout from the RNN hidden activity space, thus allowing the RNN expressivity to be decoupled from the size of the observed neural population.  To what extent might some of the described mismatches be artifacts of the limited expressivity of RNN models whose individual units are constrained to directly match observed teacher unit activity? To be concrete, in the integration example, too few units in the student network was explained as a cause for the student network mis-identifying a line attractor motif when the teacher actually employed a feedforward chain (lines 127-132; and Figure 1b). Would a mismatch still appear by when fitting partial observations using a less-constrained architecture like LFADS?
- Minor: The intro paragraph beginning on line 34 lists a number of challenges for data-constrained models. Why is it important for us to consider all of these limitations when only the partial observability is addressed in this work? There may be an opportunity to clarify the exposition here.
- Minor: Line 66: "even under relatively ideal conditions where the input to a circuit is either perfectly known or white noise". In the white noise case, were known inputs not provided to the student networks? This sentence could benefit from clarification.

**Limitations:**

The authors did not include an explicit "Limitations" section, and beyond a brief sentence in the author checklist, limitations of the techniques are not discussed explicitly in the manuscript. I would encourage the authors to consider stating some of the limitations of their studies and the particular model class chosen for the student RNNs.

---

> ### Author Rebuttal · Authors · 2024-08-06
>
> We thank the reviewer for their appreciation of our work. Below we address some of the concerns and questions that they raise:
> #### Weaknesses:
>  * We thank the reviewer for pointing out the work of Pandarinath's group, which we unfortunately missed citing in our initial manuscript. We will add the suggested references.
> #### Questions:
> * While the analytically tractable setup we describe in Section 3 is indeed constrained to a one-to-one mapping from units in the student and units in the teacher, we would like to emphasize that the motivating experiment described in Section 2 was actually performed with a latent variable model (LDS) that does not have such a constraint (See lines 109-113). Further, for our results on low-rank non-normal teacher dynamics (Section 3.1), we showed experiments in a supplementary figure that found that the spurious slow directions we derived analytically for the one-to-one mapping setup still occurred in LDS models (Supplementary F.5). We did not experiment with LFADS because it is a spiking model, whereas we focus on rate-based models throughout. Moreover, to the best of our knowledge, there's not a straightforward way of extracting spectral information from LFADS weights to make claims about attractor dynamics, whereas this can and has been done for LDS.
> * We describe the limitations of data-constrained models in detail here to help frame why the insights that are derived from such models are often rather course-grained (attractor-like properties rather than individual synaptic weights). Further, it motivates our analytical setup in Section 2, which includes two of the four stated limitations (partial observation and neuronal noise), and justifies why we focus on recovery of spectral properties over other possible criteria (e.g., weight recovery). Lastly, the discussion of these limitations supports the overall message that even in a best-case scenario where the other two limitations are ignored, data-constrained RNNs are inductively biased towards fitting (possibly spurious) attractor-like solutions.
> * In the white noise case, the students were only driven by noise; there are no known additional inputs. We will reword that sentence to make this distinction clearer.

---

> > ### Comment · Reviewer_VFVu · 2024-08-12
> >
> > Thank you for these responses. I stand by my original evaluation. I wish the authors all of the best with this work.

---

> > > ### Author Response · Authors · 2024-08-12
> > >
> > > Thank you again for your valuable and positive feedback, which has helped us improve our manuscript!

---

### Official Review · Reviewer_njnf · 2024-07-21

**Soundness:** 4
**Presentation:** 4
**Contribution:** 3
**Rating:** 6
**Confidence:** 4

**Summary:**

Deriving a mechanistic understanding of neural circuits from observations (neural recordings) is a fundamentally ill-posed problem. This paper explores and exposes this issue in controlled theoretical settings. Specifically, the authors focus on two aspects: the intrinsic biases of data-constrained surrogate models and the partial observability of the data. They demonstrate these pitfalls in both linear and non-linear systems.

**Strengths:**

This is a beautifully written paper, clear and concise. The examples are well motivated. The theoretical and experimental analyses are mathematically tight. Moreover, the issue of fitting models to partially-observed data and deriving mechanistic insights from such surrogate models is a pertinent one for neuroscience today, given the developments in recording techniques and data-driven modeling.

**Weaknesses:**

While the problem formulation is intuitive and well-motivated, the settings considered here are exceptionally constrained. This paper would greatly benefit from and be practically more useful if the authors provided scenarios where observed data has more than one plausible mechanistic explanation and how one would go about falsifying each hypothesis. The authors allude to perturbation analyses as a solution, but once again, the interpretations are unclear when we lack a "ground truth."

Are there existing biological neural datasets that the authors can use to expose possible misinterpretations of circuit mechanisms?

This question of system identification has been of long-standing interest in neuroscience. Coupled with surrogate modeling, there is a big literature on simulation-based inference and adjacent topics that would be worth adding as discussion in this current manuscript. Some representative references:

Flexible and efficient simulation-based inference for models of decision-making. Boelts et al. (2022)

Fast Inference of Spinal Neuromodulation for Motor Control using Amortized Neural Networks. Govindarajan et al. (2022)

The frontier of simulation-based inference. Cranmer et al. (2020)

**Questions:**

Please refer to the weaknesses section above.

**Limitations:**

Please refer to the weaknesses section above.

---

> ### Author Rebuttal · Authors · 2024-08-06
>
> We thank the reviewer for their careful reading of our manuscript, and are grateful for their positive assessment. Below we respond to their comments on the Weakenesses of our work:
>
> * We are glad the reviewer found our formulation of the problem intuitive, and agree that the settings we consider are constrained. However, we would argue that they are not exceptionally so. In particular, we emphasize that the integrator networks in Figure 1 do in fact have multiple plausible mechanistic explanations; by examination of the activity and network outputs alone we could not discern that the example in 2a is a line attractor, while that in 2b is a feedforward chain. Given the mismatch in the fitted and ground-truth flow fields, one could experimentally distinguish these hypothesis based on perturbing the activity of the integrator neurons in this circuit. This could be accomplished, for instance, through optogenetic stimulation (see e.g. O'Shea, Duncker, et al. 2022, or very recent work by Vinograd et al. 2024 posted after the NeurIPS submission deadline). Carefully investigating when perturbations can sensitively distinguish between mechanistic hypotheses is an important topic, but lies beyond the scope of the present work.
>
> * We agree with the reviewer that it would be interesting to expose possible misinterpretations of circuit mechanisms in existing biological datasets. However, for the purpose of the present paper we prefer to maintain our focus on synthetic data with known mechanisms, so that we can map out when mismatch in model mechanism arises. This is also motivated by the fact that conclusively demonstrating misinterpretation of mechanism in biology would require experimental manipulations; as we are not in a place to carry out such experiments, for the moment we prefer to take a more conservative approach rather than suggesting particular targets for future study. We will a discussion on this broader point, to our work.
>
> * We thank the reviewer for drawing our attention to the simulation-based inference literature, and will expand the discussion section to reflect its connections with our work.

---

> > ### Comment · Reviewer_njnf · 2024-08-09
> > **acknowledging author rebuttal**
> >
> > Thanks for your response. I've read the rebuttal and comments of other reviewers. I still think this is a good paper, but the rebuttal itself does not provide any further information. If working with biological datasets is beyond the scope of this paper, I think at least a discussion of how one can even "detect" a potential mismatch in the absence of known mechanisms seems like an important addition to the study, given the framing of the paper and its relevance for neuroscience.
> >
> > I'm keeping my original evaluation. Good luck to the authors!

---

> > > ### Author Response · Authors · 2024-08-12
> > >
> > > Thank you again for your careful evaluation! We will add a discussion of potential approaches to detecting mismatch to our manuscript, and certainly intend to follow up on this question in depth in future work.

---

### Author Rebuttal · Authors · 2024-08-06

We thank the referees for their careful reading our our manuscript, and are gratified by their positive appraisal of our work.

Here, we would like to address two common points of concern.

## Title and Framing

First, we would like to clarify why we frame the title and abstract to focus on partial observation rather than  dimensionality mismatch in general:

Partial observation is a near-universal property of data-constrained models in neuroscience, whereas restricted latent dimensionalities only apply to a subset of such models (many methods exist for fitting one-to-one mappings between student units and observed teacher units, such as FORCE, BPTT, and the MAP inference procedure we describe). Our latest experiments, the quoted reasoning of lines 127-132, and the second paragraph of the Discussion all suggest that dimensionality mismatches arising from restricted latent dimensions also contribute to the observed mechanistic mismatches. However, our analytical results only apply to the setting without the additional latent dimensionality restriction. Extending our theoretical analysis to latent variable models is an important objective of our future work. Consequently, to reflect our contributions accurately, we focus on partial observation in the title and abstract, using wording that does not rule out other possible sources of mismatch. We reiterate that we do discuss restricted latent dimensions as an additional driver of bias towards attractor-like solutions in the Discussion (paragraph 2). We nonetheless agree that this point should be expanded on, and will discuss it further in the context of our latest experiments.

## Solutions

Second, we agree with the reviewers that it is important to provide an avenue to resolve the issues we highlight in our work. As we mentioned in our manuscript and discuss in our reply to Reviewer njnf, one approach to validating putative attractor structure is to examine responses to perturbations of neural activity, for instance using optogenetics. However, we believe that it is important to focus the present manuscript on documenting the space of failure modes before we can formulate a useful proposal as to how they can be addressed. In our revised manuscript, we will expand our discussion of possible approaches using targeted optogenetic perturbations, and caveats thereof.

---

### Decision · Program_Chairs · 2024-09-25

**Decision:**

Accept (poster)

**Comment:**

This paper theoretically and numerically studies the implications of only partial observations for inferring dynamical systems models from neuroscience data. Theoretical analysis with linear models reveals that with non-normal dynamics certain mechanistic scenarios cannot be discerned under partial observations, or are wrongly identified. Numerical analysis with nonlinear models reveals further issues with spurious attractor structure. All referees felt this is an important contribution in the field of neuroscience, where dynamical systems methods are widely used yet practitioners are often not (sufficiently) aware of such issues, and technically the paper appears to be solid.

Personally, I admittedly had more reservations with this work, but will go with the referees in my vote. Perhaps in their revision the authors may want to consider the following points:

1) Partial observations are a standard problem in any empirical setting that has been widely discussed in the nonlinear dynamics and physics literature (simple google search). Different solutions for it were advanced, from simple preprocessing by optimal delay embedding (which often already does a fair job) to reconstructing unobserved variables in the model’s latent space (a problem of course if one starts with too few latent states to begin with, as apparently much neuroscience work does). Personally, I found it a bit surprising that hardly any of this literature was discussed. Also, whether simple delay embedding could already alleviate the problem in this neuroscience setting would have been an interesting piece of information for practitioners (and simple to do).

2) Another thing the authors may want to check is whether optimal regularization parameters were chosen for the reconstruction methods. This was somewhat unclear, also why different settings were chosen for different methods, and whether other adjustments of the regularization may substantially improve the recovery of the observed dynamics.